# Stagewise Training Accelerates Convergence of Testing Error Over SGD

**Zhuoning Yuan[†], Yan Yan[†], Rong Jin[‡], Tianbao Yang[†]**
[†]Department of Computer Science, The University of Iowa, Iowa City, IA 52242, USA
[‡]Machine Intelligence Technology, Alibaba Group, Bellevue, WA 98004, USA
{zhuoning-yuan, yan-yan-2, tianbao-yang}@uiowa.edu, jinrong.jr@alibaba-inc.com

## Abstract

Stagewise training strategy is widely used for learning neural networks, which runs a stochastic algorithm (e.g., SGD) starting with a relatively large step size (aka learning rate) and geometrically decreasing the step size after a number of iterations. It has been observed that the stagewise SGD has much faster convergence than the vanilla SGD with a polynomially decaying step size in terms of both training error and testing error. *But how to explain this phenomenon has been largely ignored by existing studies.* This paper provides some theoretical evidence for explaining this faster convergence. In particular, we consider a stagewise training strategy for minimizing empirical risk that satisfies the Polyak-Łojasiewicz (PL) condition, which has been observed/proved for neural networks and also holds for a broad family of convex functions. For convex loss functions and two classes of "nice-behaved" non-convex objectives that are close to a convex function, we establish faster convergence of stagewise training than the vanilla SGD under the PL condition on both training error and testing error. Experiments on stagewise learning of deep neural networks exhibits that it satisfies one type of non-convexity assumption and therefore can be explained by our theory.

## 1 Introduction

In this paper, we consider learning a predictive model by using a stochastic algorithm to minimize the expected risk via solving the following empirical risk problem:

$$\min_{\mathbf{w}\in\Omega} F_{\mathcal{S}}(\mathbf{w}) := \frac{1}{n}\sum_{i=1}^{n} f(\mathbf{w}, \mathbf{z}_i), \tag{1}$$

where $f(\mathbf{w}, \mathbf{z})$ is a smooth loss function of the model $\mathbf{w}$ on the data $\mathbf{z}$, $\Omega$ is a closed convex set, and $\mathcal{S} = \{\mathbf{z}_1, \ldots, \mathbf{z}_n\}$ denotes a set of $n$ observed data points that are sampled from an underlying distribution $\mathbb{P}_z$ with support on $\mathcal{Z}$.

There are tremendous studies devoted to solving this empirical risk minimization (ERM) problem in machine learning and related fields. Among all existing algorithms, stochastic gradient descent (SGD) is probably the simplest and attracts most attention, which takes the following update:

$$\mathbf{w}_{t+1} = \Pi_{\Omega}[\mathbf{w}_t - \eta_t \nabla f(\mathbf{w}_t, \mathbf{z}_{i_t})], \tag{2}$$

where $i_t \in \{1, \ldots, n\}$ is randomly sampled, $\Pi_{\Omega}$ is the projection operator, and $\eta_t$ is the step size that is usually decreasing to 0. Convergence theories have been extensively studied for SGD with a polynomially decaying step size (e.g., $1/t$, $1/\sqrt{t}$) for an objective that satisfies various assumptions, e.g., convexity [25], non-convexity [12], strong convexity [15], local strong convexity [27], Polyak-Łojasiewicz inequality [18], Kurdyka-Łojasiewicz inequality [30], etc. The list of papers about SGD is so long that can not be exhausted here.

The success of deep learning is mostly driven by stochastic algorithms as simple as SGD running on big data sets [20, 17]. However, an interesting phenomenon that can be observed in practice for deep learning is that no one is actually using the vanilla SGD with a polynomially decaying step size that is well studied in theory for non-convex optimization [18, 12, 9]. Instead, a common trick used to speed up the convergence of SGD is by using a stagewise step size strategy, i.e., starting from a relatively large step size and decreasing it geometrically after a number of iterations [20, 17]. Not only the convergence of training error is accelerated but also is the convergence of testing error. However, there is still a lack of theory for explaining this phenomenon. Although a stagewise step size strategy has been considered in some studies [16, 30, 18, 19, 7], none of them explains the benefit of stagewise training used in practice compared with standard SGD with a decreasing step size, especially on the convergence of testing error for non-convex problems.

**Our Contributions.** This paper aims to provide some theoretical evidence to show that an appropriate stagewise training algorithm can have faster convergence than SGD with a polynomially deccaying step size under some condition. In particular, we analyze a stagewise training algorithm under the Polyak-Łojasiewicz condition [26]:

$$2\mu(F_{\mathcal{S}}(\mathbf{w}) - \min_{\mathbf{w}\in\Omega} F_{\mathcal{S}}(\mathbf{w})) \leq \|\nabla F_{\mathcal{S}}(\mathbf{w})\|^2,$$

where $\mu$ is a constant. This property has been recently observed/proved for learning deep and shallow neural networks [13, 29, 24, 31, 5], and it also holds for a broad family of convex functions [30]. We will focus on the scenario that $\mu$ is a **small positive value** and $n$ is large, which corresponds to ill-conditioned big data problems and is indeed the case for many problems [13, 5]. We compare with two popular vanilla SGD variants with $\Theta(1/t)$ or $\Theta(1/\sqrt{t})$ step size scheme for both the convex loss and two classes of non-convex objectives that are close to a convex function. We show that the considered stagewise training algorithm has a better dependence on $\mu$ than the vanilla SGD with $\Theta(1/t)$ step size scheme for both the training error (under the same number of iterations) and the testing error (under the same number of data and a less number of iterations), while keeping the same dependence on the number of data for the testing error bound. Additionally, it has faster convergence and a smaller testing error bound than the vanilla SGD with $\Theta(1/\sqrt{t})$ step size scheme for big data.

To be fair for comparison between two algorithms, we adopt a unified approach that considers both the optimization error and the generalization error, which together with algorithm-independent optimal empirical risk constitute the testing error. In addition, we use the same tool for analysis of the generalization error - a key component in the testing error. The techniques for us to prove the convergence of optimization error and testing error are simple and standard. It is of great interest to us that simple analysis of the widely used learning strategy can possibly explain its greater success in practice than using the standard SGD method with a polynomially decaying step size.

Besides theoretical contributions, the considered algorithm also has additional features that come with theoretical guarantee for the considered non-convex problems and help improve the generalization performance, including allowing for explicit algorithmic regularization at each stage, using an averaged solution for restarting, and returning the last stagewise solution as the final solution. It is also notable that the widely used stagewise SGD is covered by the proposed framework. We refer to the considered algorithm as **stagewise regularized training algorithm or START**.

**Other closely related works.** It is notable that many papers have proposed and analyzed deterministic/stochastic optimization algorithms under the PL condition, e.g., [18, 23, 28, 2]. This list could be long if we consider its equivalent condition in the convex case. However, none of them exhibits the benefit of stagewise learning strategy used in practice. One may also notice that linear convergence for the optimization error was proved for a stochastic variance reduction gradient method [28]. Nevertheless, its uniform stability bound remains unclear for making a fair comparison with the considered algorithms in this paper, and variance reduction method is not widely used for deep learning.

We also notice that some recent studies [21, 32, 5] have used other techniques (e.g., data-dependent bound, average stability, point-wise stability) to analyze the generalization error of a stochastic algorithm. Nevertheless, we believe similar techniques can be also used for analyzing stagewise learning algorithm, which is beyond the scope of this paper. The generalization error results in [32, 5] under PL condition are not directly comparable to ours because they have stronger assumptions (e.g., the global minimizer is unique for deriving uniform stability [5], the condition number is small [32]). Finally, it was brought to our attention when a preliminary version of this paper is done that an independent work [11] observes a similar advantage of stagewise SGD over SGD with a polynomially

decaying step size lying at the better dependence on the condition number. However, they only analyze the strongly convex quadratic case and the training error of ERM.

## 2  Preliminaries and Notations

Let $\mathcal{A}$ denote a randomized algorithm, which returns a randomized solution $\mathbf{w}_{\mathcal{S}} = \mathcal{A}(\mathcal{S})$ based on the given data set $\mathcal{S}$. Denote by $\mathrm{E}_{\mathcal{A}}$ the expectation over the randomness in the algorithm and by $\mathrm{E}_{\mathcal{S}}$ expectation over the randomness in the data set. When it is clear from the context, we will omit the subscript $\mathcal{S}$ and $\mathcal{A}$ in the expectation notations. Let $\mathbf{w}_{\mathcal{S}}^* \in \arg\min_{\mathbf{w} \in \Omega} F_{\mathcal{S}}(\mathbf{w})$ denote an empirical risk minimizer, and $F(\mathbf{w}) = \mathrm{E}_{\mathbf{z}}[f(\mathbf{w}, \mathbf{z})]$ denote the true risk of $\mathbf{w}$ (also called **testing error** in this paper). We use $\|\cdot\|$ to denote the Euclidean norm, and use $[n] = \{1, \ldots, n\}$.

In order to analyze the testing error convergence of a random solution, we use the following decomposition of testing error.

$$\mathrm{E}_{\mathcal{A},\mathcal{S}}[F(\mathbf{w}_{\mathcal{S}})] = \mathrm{E}_{\mathcal{S}}[F_{\mathcal{S}}(\mathbf{w}_{\mathcal{S}}^*)] + \mathrm{E}_{\mathcal{S}}\underbrace{\mathrm{E}_{\mathcal{A}}[F_{\mathcal{S}}(\mathbf{w}_{\mathcal{S}}) - F_{\mathcal{S}}(\mathbf{w}_{\mathcal{S}}^*)]}_{\varepsilon_{opt}} + \underbrace{\mathrm{E}_{\mathcal{A},\mathcal{S}}[F(\mathbf{w}_{\mathcal{S}}) - F_{\mathcal{S}}(\mathbf{w}_{S})]}_{\varepsilon_{gen}},$$

where $\varepsilon_{opt}$ measures the optimization error, i.e., the difference between empirical risk (or called **training error**) of the returned solution $\mathbf{w}_{\mathcal{S}}$ and the optimal value of the empirical risk, and $\varepsilon_{gen}$ measures the generalization error, i.e., the difference between the true risk of the returned solution and the empirical risk of the returned solution. The difference $\mathrm{E}_{\mathcal{A},\mathcal{S}}[F(\mathbf{w}_{\mathcal{S}})] - \mathrm{E}_{\mathcal{S}}[F_{\mathcal{S}}(\mathbf{w}_{\mathcal{S}}^*)]$ is an upper bound of the so-called **excess risk bound** in the literature, which is defined as $\mathrm{E}_{\mathcal{A},\mathcal{S}}[F(\mathbf{w}_{\mathcal{S}})] - \min_{\mathbf{w} \in \Omega} F(\mathbf{w})$. It is notable that the first term $\mathrm{E}_{\mathcal{S}}[F_{\mathcal{S}}(\mathbf{w}_{\mathcal{S}}^*)]$ in the above bound is independent of the choice of randomized algorithms. Hence, in order to compare the performance of different randomized algorithms, we can focus on analyzing $\varepsilon_{opt}$ and $\varepsilon_{gen}$. For analyzing the generalization error, we will leverage the uniform stability tool [4]. A randomized algorithm $\mathcal{A}$ is called $\epsilon$-uniformly stable if for all data sets $\mathcal{S}, \mathcal{S}' \in \mathcal{Z}^n$ that differs at most one example the following holds:

$$\varepsilon_{stab} := \sup_{\mathbf{z}} \mathrm{E}_{\mathcal{A}}[f(\mathcal{A}(\mathcal{S}), \mathbf{z}) - f(\mathcal{A}(\mathcal{S}'), \mathbf{z})] \le \epsilon.$$

A well-known result is that if $\mathcal{A}$ is $\epsilon$-uniformly stable, then its generalization error is bounded by $\epsilon$ [4], i.e., if $\mathcal{A}$ is $\epsilon$-uniformly stable, we have $\varepsilon_{gen} \le \epsilon$. In light of the above discussion, in order to compare the convergence of testing error of different randomized algorithms, it suffices to analyze their convergence in terms of optimization error and their uniform stability. We would like to emphasize that the PL condition is not used in our generalization error analysis by uniform stability, which makes our comparison to the results in [14] fair.

A function $f(\mathbf{w})$ is $L$-smooth if it is differentiable and its gradient is $L$-Lipchitz continuous, i.e., $\|\nabla f(\mathbf{w}) - \nabla f(\mathbf{u})\| \le L\|\mathbf{w} - \mathbf{u}\|, \forall \mathbf{w}, \mathbf{u} \in \Omega$. A function $f(\mathbf{w})$ is $G$-Lipchitz continuous if $\|\nabla f(\mathbf{w})\| \le G, \forall \mathbf{w} \in \Omega$. We summarize the used assumptions below with some positive $L, \sigma, G, \mu$ and $\epsilon_0$.

**Assumption 1.** *Assume that*

  *(i)  $f(\mathbf{w}, \mathbf{z})$ is $L$-smooth in terms of $\mathbf{w} \in \Omega$ for every $\mathbf{z} \in \mathcal{Z}$.*

  *(ii)  $f(\mathbf{w}, \mathbf{z})$ is finite-valued and $G$-Lipchitz continuous in terms of $\mathbf{w} \in \Omega$ for every $\mathbf{z} \in \mathcal{Z}$.*

  *(iii)  there exists $\sigma$ such that $\mathrm{E}_i[\|\nabla f(\mathbf{w}, \mathbf{z}) - \nabla F_{\mathcal{S}}(\mathbf{w})\|^2] \le \sigma^2$ for $\mathbf{w} \in \Omega$.*

  *(iv)  $F_{\mathcal{S}}(\mathbf{w})$ satisfies the PL condition for any $S$ of size $n$, i.e., there exists $\mu$*

$$2\mu(F_{\mathcal{S}}(\mathbf{w}) - F_{\mathcal{S}}(\mathbf{w}_{\mathcal{S}}^*)) \le \|\nabla F_{\mathcal{S}}(\mathbf{w})\|^2, \forall \mathbf{w} \in \Omega.$$

  *(v)  For an initial solution $\mathbf{w}_0 \in \Omega$, there exists $\epsilon_0$ such that $F_{\mathcal{S}}(\mathbf{w}_0) - F_{\mathcal{S}}(\mathbf{w}_{\mathcal{S}}^*) \le \epsilon_0$.*

**Remark 1:** The second assumption is imposed for the analysis of uniform stability of a randomized algorithm. W.o.l.g we assume $|f(\mathbf{w}, \mathbf{z})| \le 1, \forall \mathbf{w} \in \Omega$. The third assumption is for the purpose of analyzing optimization error. It is notable that $\sigma^2 \le 4G^2$. For simplicity, we assume the PL condition of $F_{\mathcal{S}}(\mathbf{w})$ holds uniformly over $\mathcal{S}$. It is known that the PL condition is much weaker than strong convexity. If $F_{\mathcal{S}}$ is strongly convex, $\mu$ corresponds to the strong convexity parameter. In this paper, we are particularly interested in the case when $\mu$ is small, i.e. the condition number $L/\mu$ is large.

**Remark 2:** It is worth mentioning that we do not assume the PL condition holds in the whole space $\mathbb{R}^d$. Hence, our analysis presented below can capture some cases that the PL condition only holds in a local space $\Omega$ that contains a global minimum. For example, the recent papers by [10, 29, 1] shows that the global minimum of learning a two-layer and deep overparameterized neural networks resides in a ball centered around a random initial solution and the PL condition holds in the ball.

# 3 Review: SGD under PL Condition

In this section, we review the training error convergence and generalization error of SGD with a decreasing step size for functions satisfying the PL condition in order to derive its testing error bound. We will focus on SGD using the step size $\Theta(1/t)$ and briefly mention the results corresponding to $\Theta(1/\sqrt{t})$ at the end of this section. We would like to emphasize the results presented in this section are mainly from existing works [18, 14]. The optimization error and the uniform stability of SGD have been studied in these two papers separately. Since we are not aware of any studies that piece them together, it is of our interest to summarize these results here for comparing with our new results established later in this paper. Let us first consider the optimization error convergence, which has been analyzed in [18] and is summarized below.

**Theorem 1.** *[18] Suppose $\Omega = \mathbb{R}^d$. Under Assumption 1 (i), (iv) and $\mathrm{E}_i[\|\nabla f(\mathbf{w}, \mathbf{z}_i)\|^2] \leq G^2$, by setting $\eta_t = \frac{2t+1}{2\mu(t+1)^2}$ in the update of SGD (2), we have*

$$\mathrm{E}[F_{\mathcal{S}}(\mathbf{w}_T) - F_{\mathcal{S}}(\mathbf{w}_{\mathcal{S}}^*)] \leq \frac{LG^2}{2T\mu^2}, \tag{3}$$

*and by setting $\eta_t = \eta$, we have $\mathrm{E}[F_{\mathcal{S}}(\mathbf{w}_T) - F_{\mathcal{S}}(\mathbf{w}_{\mathcal{S}}^*)] \leq (1 - 2\eta\mu)^T (F_{\mathcal{S}}(\mathbf{w}_0) - F_{\mathcal{S}}(\mathbf{w}_{\mathcal{S}}^*)) + \frac{\eta LG^2}{4\mu}$.*

**Remark 3:** In order to have an $\epsilon$ optimization error, one can set $T = \frac{LG^2}{2\mu^2\epsilon}$ in the decreasing step size setting. In the constant step size setting, one can set $\eta = \frac{2\mu\epsilon}{LG^2}$ and $T = \frac{LG^2}{4\mu^2\epsilon} \log(2\epsilon_0/\epsilon)$, where $\epsilon_0 \geq F_{\mathcal{S}}(\mathbf{w}_0) - F_{\mathcal{S}}(\mathbf{w}_{\mathcal{S}}^*)$ is the initial optimization error bound. [18] also mentioned a stagewise step size strategy based on the second result above. By starting with $\eta_1 = \frac{\epsilon_0\mu}{LG^2}$ and running for $t_1 = \frac{LG^2 \log 4}{2\mu^2\epsilon_0}$ iterations, and restarting the second stage with $\eta_2 = \eta_1/2$ and $t_2 = 2t_1$, then after $K = \log(\epsilon_0/\epsilon)$ stages, we have optimization error less than $\epsilon$, and the total iteration complexity is $O(\frac{LG^2 \log 4}{\mu^2\epsilon})$. We can see that the analysis of [18] cannot explain why stagewise optimization strategy brings any improvement compared with SGD with a decreasing step size of $O(1/t)$. No matter which step size strategy is used among the ones discussed above, the total iteration complexity is $O(\frac{L}{\mu^2\epsilon})$. It is also interesting to know that the above convergence result does not require the convexity of $f(\mathbf{w}, \mathbf{z})$. On the other hand, it is unclear how to directly analyze SGD with a polynomially decaying step size for a convex loss to obtain a better convergence rate than (3).

The generalization error bound by uniform stability for both convex and non-convex losses have been analyzed in [14]. We just need to plug the step size of SGD in Theorem 1 into their results (Theorem 3.7 and Theorem 3.8) to prove the uniform stability. For the sake of space, we summarize the uniform stability results in Theorem 11 in the supplement. Combining the optimization error and uniform stability, we obtain the convergence of testing error of SGD for smooth loss functions under the PL condition. By optimizing the value of $T$ in the bounds, we obtain the following testing error bound dependent on $n$ only.

**Theorem 2.** *Suppose Assumption 1 holds. If $f(\cdot, \mathbf{z})$ is convex for any $\mathbf{z} \in \mathcal{Z}$, with step size $\eta_t = \frac{2t+1}{2\mu(t+1)^2}$ and $T = \frac{nLG^2}{4(L+2G^2)\mu}$ iterations SGD returns a solution $\mathbf{w}_T$ satisfying*

$$\mathrm{E}_{\mathcal{A},\mathcal{S}}[F(\mathbf{w}_T)] \leq \mathrm{E}_{\mathcal{S}}[F_{\mathcal{S}}(\mathbf{w}_{\mathcal{S}}^*)] + \frac{2(L+2G^2)}{n\mu} + \frac{(L+2G^2)\log(T+1)}{n\mu}.$$

*If $f(\cdot, \mathbf{z})$ is non-convex for any $\mathbf{z} \in \mathcal{Z}$, with step size $\eta_t = \frac{2t+1}{2\mu(t+1)^2}$ and $T = \max\{\frac{\sqrt{(n-1)LG}}{\sqrt{8}\mu^{3/4}}, \frac{\sqrt{(n-1)LG}}{2\mu e\hat{G}}\}$ iterations SGD returns a solution $\mathbf{w}_T$ satisfying*

$$\mathrm{E}_{\mathcal{A},\mathcal{S}}[F(\mathbf{w}_T)] \leq \mathrm{E}_{\mathcal{S}}[F_{\mathcal{S}}(\mathbf{w}_{\mathcal{S}}^*)] + 2\min\left\{\frac{\sqrt{2}L^{1/2}G^{3/2}}{\sqrt{(n-1)}\mu^{5/4}}, \frac{\sqrt{Le^{2\hat{G}}}G}{\sqrt{n-1}\mu}\right\}.$$

**Remark 4:** If the loss is convex, the excess risk bound is in the order of $O(\frac{L\log(nL/\mu)}{n\mu})$ by running SGD with $T = O(nL/\mu)$ iterations. It notable that an $O(1/n)$ excess risk bound is called the fast rate in the literature. If the loss is non-convex and $2G/\sqrt{\mu} > e^{2\hat{G}}$ (an interesting case [1]), the excess risk bound is in the order of $O(\frac{\sqrt{L}}{\sqrt{n}\mu})$ by running SGD with $T = O(\sqrt{nL}/\mu)$ iterations. When $\mu$ is

| **Algorithm 1** START Algorithm: $\text{START}(F_{\mathcal{S}}, \mathbf{w}_0, \gamma, K)$ | **Algorithm 2** $\text{SGD}(F_{\mathbf{w}_1}^{\gamma}, \mathbf{w}_1, \eta, T)$ |
|---|---|
| 1: **Input:** $\mathbf{w}_0$, $\gamma$ and $K$ | 1: **for** $t = 1, \dots, T$ **do** |
| 2: **for** $k = 1, \dots, K$ **do** | 2:   Sample a random data $\mathbf{z}_{i_t} \in \mathcal{S}$ |
| 3:   Let $F_{\mathbf{w}_{k-1}}^{\gamma}(\mathbf{w}) = F_{\mathcal{S}}(\mathbf{w}) + \frac{1}{2\gamma}\|\mathbf{w} - \mathbf{w}_{k-1}\|^2$ | 3:   $\mathbf{w}_{t+1} = \min_{\mathbf{w} \in \Omega} \nabla f(\mathbf{w}_t, \mathbf{z}_{i_t})^{\top}\mathbf{w} +$ |
| 4:   $\mathbf{w}_k = \text{SGD}(F_{\mathbf{w}_{k-1}}^{\gamma}, \mathbf{w}_{k-1}, \eta_k, T_k)$ | $\quad \frac{1}{2\eta}\|\mathbf{w} - \mathbf{w}_t\|^2 + \frac{1}{2\gamma}\|\mathbf{w} - \mathbf{w}_1\|^2$ |
| 5: **end for** | 4: **end for** |
| 6: **Return:** $\mathbf{w}_K$ | 5: **Output**: $\widehat{\mathbf{w}}_T = \mathcal{O}(\mathbf{w}_1, \dots, \mathbf{w}_{T+1})$ |

very small, the convergence of testing error is very slow. In addition, the number of iterations is also scaled by $1/\mu$ for achieving a minimal excess risk bound.

**Remark 5:** Another possible choice of decreasing step size is $O(1/\sqrt{t})$, which yields an $O(1/\sqrt{T})$ convergence rate for $F_{\mathcal{S}}(\widehat{\mathbf{w}}_T) - F_{\mathcal{S}}(\mathbf{w}_{\mathcal{S}}^*)$ in the convex case [25] or for $\|\nabla F_{\mathcal{S}}(\mathbf{w}_t)\|^2$ in the non-convex case with a randomly sampled $t$ [12]. In the latter case, it also implies a worse convergence rate of $O(1/(\mu\sqrt{T}))$ for the optimization error $F_{\mathcal{S}}(\mathbf{w}_t) - \min_{\mathbf{w}} F_{\mathcal{S}}(\mathbf{w})$ under the PL condition [2]. Regarding the uniform stability, the step size of $O(1/\sqrt{t})$ will also yield a worse growth rate in terms of $T$ [14]. For example, if the loss function is convex, the generalization error by uniform stability scales as $O(\sqrt{T}/n)$ and hence the testing error bound is in the order of $O(1/\sqrt{n\mu})$, which is worse than the above testing error bound $\widetilde{O}(1/(n\mu))$ for the big data setting $\mu \geq \Omega(1/n)$. Hence, below we will focus on the comparison with the theoretical results in Theorem 2.

## 4   START for a Convex Function

First, let us present the algorithm that we intend to analyze in Algorithm 1. At the $k$-th stage, a regularized funciton $F_{\mathbf{w}_{k-1}}^{\gamma}(\mathbf{w})$ is constructed that consists of the original objective $F_{\mathcal{S}}(\mathbf{w})$ and a quadratic regularizer $\frac{1}{2\gamma}\|\mathbf{w} - \mathbf{w}_{k-1}\|^2$. The reference point $\mathbf{w}_{k-1}$ is a returned solution from the previous stage, which is also used for an initial solution for the current stage. Adding the strongly convex regularizer at each stage is not essential but could be helpful for reducing the generalization error and is also important for one class of non-convex loss considered in next section. For each regularized problem, SGD with a constant step size is employed for a number of iterations with an appropriate returned solution. We will reveal the value of step size, the number of iterations and the returned solution for each class of problems separately. Note that the widely used stagewise SGD falls into the framework of START when $\gamma = \infty$ and $\mathcal{O} = (\mathbf{w}_1, \dots, \mathbf{w}_{T+1}) = \mathbf{w}_{T+1}$.

In this section, we will analyze START algorithm for a convex function under the PL condition. We would like to point out that similar algorithms have been proposed and analyzed in [16, 30] for convex problems. They focus on analyzing the convergence of optimization error for convex problems under a quadratic growth condition or more general local error bound condition. In the following, we will show that the PL condition implies a quadratic growth condition. Hence, their algorithms can be used for optimizing $F_{\mathcal{S}}$ as well enjoying a similar convergence rate in terms of optimization error. However, there is still slight difference between the analyzed algorithm from their considered algorithms. In particular, the regularization term $\frac{1}{2\gamma}\|\mathbf{w} - \mathbf{w}_{k-1}\|^2$ is absent in [16], which corresponds to $\gamma = \infty$ in our case. However, adding a small regularization (with not too large $\gamma$) can possibly help reduce the generalization error. In addition, their initial step size is scaled by $1/\mu$. The initial step size of our algorithm depends on the quality of initial solution that seems more natural and practical. A similar regularization at each stage is also used in [30]. But their algorithm will suffer from a large generalization error, which is due to the key difference between START and their algorithm (ASSG-r). In particular, they use a geometrically decreasing the parameter $\gamma_k$ starting from a relatively large value in the order of $O(1/(\mu\epsilon))$ with a total iteration number $T = O(1/(\mu\epsilon))$. According to our analysis of generalization error (see Theorem 4), their algorithm has a generalization error in the order of $O(T/n)$ in contrast to $\log T/n$ of our algorithm.

Below, we summarize the convergence of optimization error in Theorem 3 and generalization error in Theorem 4. We need the following lemma for the optimization error analysis.

**Lemma 1.** *If $F_\mathcal{S}(\mathbf{w})$ satisfies the PL condition, then for any $\mathbf{w} \in \Omega$ we have*

$$\|\mathbf{w} - \mathbf{w}_\mathcal{S}^*\|^2 \leq \frac{1}{2\mu}(F_\mathcal{S}(\mathbf{w}) - F_\mathcal{S}(\mathbf{w}_\mathcal{S}^*)), \tag{4}$$

*where $\mathbf{w}_\mathcal{S}^*$ is the closest optimal solution to $\mathbf{w}$.*

**Remark 6:** The above result does not require the convexity of $F_\mathcal{S}$. For a proof, please refer to [3, 18]. Indeed, this error bound condition instead of the PL condition is enough to derive the results in Section 4 and Section 5.

Below, we let $\mathbf{w}_t^k$ denote the solution computed during the $k$-th stage at the $t$-th iteration.

**Theorem 3.** *(Optimization Error) Suppose Assumption 1 holds, and $f(\mathbf{w}, \mathbf{z})$ is a convex function of $\mathbf{w}$. Then by setting $\gamma \geq 1.5/\mu$ and $T_k = \frac{9\sigma^2}{2\mu\epsilon_k\alpha}, \eta_k = \frac{\epsilon_k\alpha}{3\sigma^2}, \mathcal{O}(\mathbf{w}_1^k, \ldots, \mathbf{w}_{T_k+1}^k) = \sum_{t=1}^{T_k} \mathbf{w}_{t+1}^k/T_k$, where $\epsilon_k = \epsilon_0/2^k$, $\alpha \leq \min(1, \frac{3\sigma^2}{\epsilon_0 L})$, after $K = \log(\epsilon_0/\epsilon)$ stages with a total iteration complexity of $O(L/(\mu\epsilon))$ we have $\mathrm{E}[F_\mathcal{S}(\mathbf{w}_K) - F_\mathcal{S}(\mathbf{w}_\mathcal{S}^*)] \leq \epsilon$.*

**Remark 7:** Compared to the result in Theorem 1, the convergence rate of START is faster by a factor of $O(1/\mu)$. It is also notable that $\gamma$ can be as large as $\infty$ in the convex case.

By showing $\sup_\mathbf{z} \mathrm{E}_\mathcal{A}[f(\mathbf{w}_K, \mathbf{z}) - f(\mathbf{w}_K', \mathbf{z})] \leq \epsilon$, we can show the generalization error is bounded by $\epsilon$, where $\mathbf{w}_K$ is learned on a data set $\mathcal{S}$ and $\mathbf{w}_K'$ is learned a different data set $\mathcal{S}'$ that only differs from $\mathcal{S}$ at most one example. Our analysis is closely following the route in [14]. The difference is that we have to consider the difference on the reference points $\mathbf{w}_{k-1}$ of two copies of our algorithm on two data sets $\mathcal{S}, \mathcal{S}'$.

**Theorem 4.** *(Uniform Stability) After $K$ stages, START satisfies uniform stability with*

$$\varepsilon_{stab} \leq \begin{cases} \frac{2\gamma G^2 \sum_{k=1}^K (1-(\frac{\gamma}{\eta_k+\gamma})^{T_k})}{n} \leq \frac{2G^2 \sum_{k=1}^K \eta_k T_k}{n} & \text{if } \gamma < \infty \\ \frac{2G^2 \sum_{k=1}^K \eta_k T_k}{n} & \text{else} \end{cases}.$$

**Put them Together.** Finally, we have the following testing error bound of $\mathbf{w}_K$ returned by START.

**Theorem 5.** *(Testing Error) After $K = \log(\epsilon_0/\epsilon)$ stages with a total number of iterations $T = \frac{18\sigma^2}{\alpha\mu\epsilon}$. The testing error of $\mathbf{w}_K$ is bounded by*

$$\mathrm{E}_{\mathcal{A},\mathcal{S}}[F(\mathbf{w}_K)] \leq \mathrm{E}[F_\mathcal{S}(\mathbf{w}_\mathcal{S}^*)] + \epsilon + \frac{3G^2 \log(\epsilon_0/\epsilon)}{n\mu}.$$

**Remark 8:** Let $\epsilon = \frac{1}{n\mu}$, the excess risk bound becomes $O(\log(n\mu)/(n\mu))$ and the total iteration complexity is $T = O(nL)$. This improves the convergence of testing error of SGD stated in Theorem 2 for the convex case when $\mu \ll 1$, which needs $T = O(nL/\mu)$ iterations and has a testing error bound of $O(L \log(nL/\mu)/(n\mu))$.

## 5 START for Non-Convex Functions

Next, we will establish faster convergence of START than SGD for "nice-behaved" non-convex functions. In particular, we will consider two classes of non-convex functions that are are close to a convex function, namely one-point weakly quasi-convex and weakly convex functions. We first introduce the definitions of these functions followed by some discussions.

**Definition 1** (One-point Weakly Quasi-Convex). *A non-convex function $F$ is called one-point $\theta$-weakly quasi-convex for $\theta > 0$ if there exists a global minimum $\mathbf{w}^*$ such that*

$$\nabla F(\mathbf{w})^\top (\mathbf{w} - \mathbf{w}^*) \geq \theta(F(\mathbf{w}) - F(\mathbf{w}^*)), \forall \mathbf{w} \in \Omega. \tag{5}$$

**Definition 2** (Weakly Convex). *A non-convex function $F$ is $\rho$-weakly convex for $\rho > 0$ if $F(\mathbf{w}) + \frac{\rho}{2}\|\mathbf{w}\|^2$ is convex.*

It is interesting to connect one-point weakly quasi-convexity to one-point strong convexity that has been considered for non-convex optimization, especially optimizing neural networks [24, 19].

**Definition 3** (One-point Strongly Convex). *A non-convex function $F$ is one-point strongly convex with respect to a global minimum $\mathbf{w}^*$ if there exists $\mu_1 > 0$ such that*

$$\nabla F(\mathbf{w})^\top (\mathbf{w} - \mathbf{w}^*) \geq \mu_1 \|\mathbf{w} - \mathbf{w}^*\|^2.$$

The following lemma shows that one-point strong convexity implies both the PL condition and the one-point weakly quasi-convexity.

**Lemma 2.** *Suppose $F$ is $L$-smooth and one-point strongly convex w.r.t $\mathbf{w}^*$ with $\mu_1 > 0$ and $\nabla F(\mathbf{w}^*) = 0$, then*

$$\min\{\|\nabla F(\mathbf{w})\|^2, \nabla F(\mathbf{w})^\top(\mathbf{w} - \mathbf{w}^*)\} \geq \frac{2\mu_1}{L}(F(\mathbf{w}) - F(\mathbf{w}^*))$$

For "nice-behaved" one-point weakly quasi-convex function $F_\mathcal{S}(\mathbf{w})$ that satisfies the PL condition, we are interested in the case that $\theta$ is a constant close to or larger than 1. Note that a convex function has $\theta = 1$ and a strongly convex function has $\theta > 1$. For the case of $\mu \ll 1$ in the PL condition, this indicates that $\nabla F(\mathbf{w})^\top(\mathbf{w} - \mathbf{w}^*)$ is larger than $\|\nabla F(\mathbf{w})\|^2$, which further implies that $\|\mathbf{w} - \mathbf{w}^*\| \geq \|\nabla F(\mathbf{w})\|$. Intuitively, this inequality (illustrated in Figure 2 in appendix) also connects itself to the flat minimum that has been observed in deep learning experiments [6]. For "nice-behaved" weakly convex function, we are interested in the case that $\rho \leq \mu/4$ is close to zero. Weakly convex functions with a small $\rho$ have been considered in the literature of non-convex optimization [22]. In both cases, we will establish faster convergence of optimization error and testing error of START.

**Convergence of Optimization Error.** The approach of optimization error analysis for the considered non-convex functions is similar to that of convex functions. We also first analyze the convergence of SGD for each stage and then extend it to $K$ stages for START. Due to limit of space, we only summarize the final convergence results here for the two classes of non-convex functions separately.

**Theorem 6.** *Suppose $F_\mathcal{S}(\mathbf{w})$ is **one-point $\theta$-weakly quasi-convex** w.r.t $\mathbf{w}_\mathcal{S}^*$ and (4) holds for the same $\mathbf{w}_\mathcal{S}^*$. Then by setting $\gamma \geq 1.5/(\theta\mu)$, $\eta_k = \frac{2\epsilon_k\theta}{3G^2}$, $T_k = \frac{9G^2}{4\mu\epsilon_k\theta^2}$ and $\mathcal{O}(\mathbf{w}_1^k, \ldots, \mathbf{w}_{T_k+1}^k) = \mathbf{w}_\tau^k$ where $\tau \in \{1, \ldots, T_k\}$ is randomly sampled, after $K = \log(\epsilon_k/\epsilon)$ stages we have $\mathrm{E}[F_\mathcal{S}(\mathbf{w}_K) - F_\mathcal{S}(\mathbf{w}_\mathcal{S}^*)] \leq \epsilon$. The total iteration complexity is $O(\frac{1}{\theta^2\mu\epsilon})$.*

**Theorem 7.** *Suppose Assumption 1 holds, and $F_\mathcal{S}(\mathbf{w})$ is $\rho$-**weakly convex** with $\rho \leq \mu/4$. Then by setting $\gamma = 4/\mu \leq 1/\rho$, $\eta_k = \frac{\epsilon_k\alpha}{4\sigma^2} \leq 1/L$, $T_k = \frac{4\sigma^2}{\mu\epsilon_k\alpha}$ and $\mathcal{O}(\mathbf{w}_1^k, \ldots, \mathbf{w}_{T_k+1}^k) = \sum_{t=1}^{T_k}\mathbf{w}_{t+1}^k/T_k$, where $\alpha \leq \min(1, \frac{2\sigma^2}{\epsilon_0 L})$, and after $K = \log(\epsilon_0/\epsilon)$ stages we have $\mathrm{E}[F_\mathcal{S}(\mathbf{w}_K) - F_\mathcal{S}(\mathbf{w}_\mathcal{S}^*)] \leq \epsilon$. The total iteration complexity is $O(\frac{1}{\alpha\mu\epsilon})$.*

**Remark 9:** Several differences are noticeable between the two classes of non-convex functions: (i) $\gamma$ in the weakly quasi-convex case can be as large as $\infty$, in contrast it is required to be smaller than $1/\rho$ in the weakly convex case; (ii) the returned solution by SGD at the end of each stage is a randomly selected solution in the weakly quasi-convex case and is an averaged solution in the weakly convex case. Finally, we note that the total iteration complexity for both cases is $O(1/\mu\epsilon)$ under $\theta \approx 1$ and $\rho \leq O(\mu)$, which is better than $O(1/\mu^2\epsilon)$ of SGD as in Theorem 1.

**Generalization Error.** The analysis of generalization error follows similarly to the non-convex case in [14]. Please note that the quasi- and weak-convexity are not directly leveraged in the generalization error analysis. The only thing that matters here is the value of step size. Hence, we present a unified result for the two cases below.

**Theorem 8.** *Let $S_{K-1} = \sum_{k=1}^{K-1} T_k = O(\frac{1}{\mu\epsilon\bar{\alpha}})$ and $\eta_K \leq c/(\mu T_K)$, where $\bar{\alpha} = \theta^2, c = 1.5/\theta$ in the one-point weakly quasi-convex case and $\bar{\alpha} = \alpha, c = 1$ in the weakly convex case. Then we have*

$$\varepsilon_{stab} \leq \frac{S_{K-1}}{n} + \frac{1 + \mu/(Lc)}{n-1}(2G^2c/\mu)^{1/(1+Lc/\mu)}T_k^{\frac{Lc/\mu}{Lc/\mu+1}}$$

By putting the optimization error and generalization error together, we have the following testing error bound.

**Theorem 9.** *Under the same assumptions as in Theorem 6 or 7 and $\mu \ll 1$. After $K = \log(\epsilon_0/\epsilon)$ stages with a total number of iterations $T = O(\frac{1}{\bar{\alpha}\mu\epsilon})$, the testing error bound of $\mathbf{w}_K$ is*

$$\mathrm{E}_{\mathcal{A},\mathcal{S}}[F(\mathbf{w}_K)] \leq \mathrm{E}[F_\mathcal{S}(\mathbf{w}_\mathcal{S}^*)] + \epsilon + O(\frac{1}{n\bar{\alpha}\mu\epsilon}).$$

**Remark 10:** We are mostly interested in the case when $\theta$ is constant close to or larger than 1. By setting $\epsilon = \Theta(1/\sqrt{n\bar{\alpha}\mu})$, we have the excess risk bounded by $O(\frac{1}{\sqrt{n\bar{\alpha}\mu}})$ under the total iteration complexity $T = O(\sqrt{n/(\bar{\alpha}\mu)})$. This improves the testing error bound of SGD stated in Theorem 2 for the non-convex case when $\mu \leq \bar{\alpha}$, which needs $T = O(\sqrt{n}/\mu)$ iterations and suffers a testing error bound of $O(1/(\sqrt{n}\mu))$.

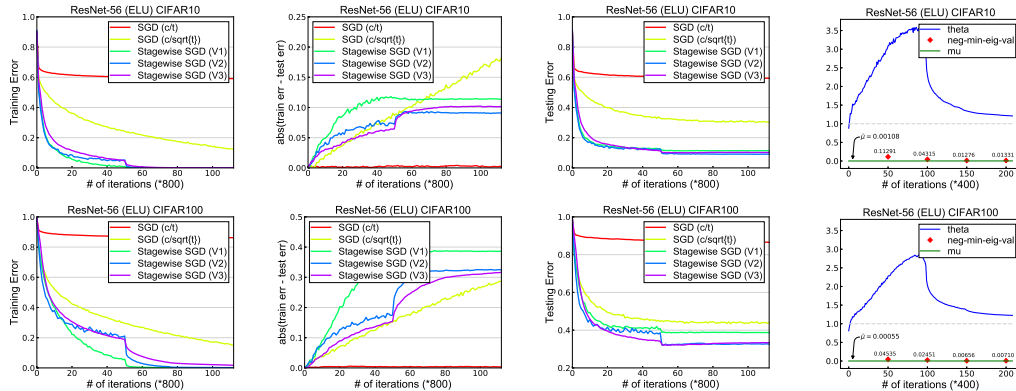

Figure 1: From left to right: training, generalization and testing error, and verifying assumptions for stagewise learning of ResNets.

Finally, it is worth noting that our analysis is applicable to an approximate optimal solution $\mathbf{w}_{\mathcal{S}}^{*}$ as long as the inequality (4) and (5) hold for that particular $\mathbf{w}_{\mathcal{S}}^{*}$. This fact is helpful for us to verify the assumptions in numerical experiments.

## 6  Numerical Experiments

We focus experiments on non-convex deep learning, and include in the supplement some experimental results of START for convex functions that satisfy the PL condition. The numerical experiments mainly serve two purposes: (i) verifying that using different algorithmic choices in practice (e.g, regularization, averaged solution) is consistent with the provided theory; (ii) verifying the assumptions made for non-convex objectives in our analysis in order to support our theory.

We compare stagewise learning with different algorithmic choices against SGD using two polynomially decaying step sizes (i.e., $O(1/t)$ and $O(1/\sqrt{t})$). For stagewise learning, we consider the widely used version that corresponds to START with $\gamma = \infty$ and the returned solution at each stage being the last solution, which is denoted as stagewise SGD (V1). We also implement other two variants of START that solves a regularized function at each stage (corresponding to $\gamma < \infty$) and uses the last solution or the averaged solution for the returned solution at each stage. We refer to these variants as stagewise SGD (V2) and (V3), respectively.

We conduct experiments on two datasets CIFAR-10, -100 using different neural network structures, including residual nets and convolutional neural nets without skip connection. Two residual nets namely ResNet20 and ResNet56 [17] are used for CIFAR-10 and CIFAR-100. For each network structure, we use two types of activation functions, namely RELU and ELU ($\alpha = 1$) [8]. ELU is smooth that is consistent with our assumption. Although RELU is non-smooth, we would like to show that the provided theory can also explain the good performance of stagewise SGD. For stagewise SGD on CIFAR datasets, we use a similar stagewise step size strategy as in [17], i.e., the step size is decreased by 10 times at 40k, 60k iterations. For all algorithms, we select the best initial step size from $10^{-3} \sim 10^{3}$ and the best regularization parameter $1/\gamma$ of stagewise SGD (V2, V3) from $0.0001 \sim 0.1$ by cross-validation based on performance on a validation data. Due to limit of space, we only report the results for using ResNet56 and ELU and no weight decay, and results for other settings are included in the supplement.

The training error, generalization error and testing error are shown in Figure 1. We can see that SGD with a decreasing step size converges slowly, especially SGD with a step size proportional to $1/t$. It is because that the initial step size of SGD ($c/t$) is selected as a small value less than 1. We observe that using a large step size cannot lead to convergence. In terms of different algorithmic choices of START, we can see that using an explicit regularization as in V2, V3 can help reduce the generalization error that is consistent with theory, but also slows down the training a little. Using an averaged solution as the returned solution in V3 can further reduce the generalization error but also further slow downs the training. Overall, stagewise SGD (V2) achieves the best tradeoff in training error convergence and generalization error, which leads to the best testing error.

Finally, we verify the assumptions about the non-convexity made in Section 5. To this end, on a selected $\mathbf{w}_t$ we compute the value of $\theta$, i.e., the ratio of $\nabla F_{\mathcal{S}}(\mathbf{w}_t)^{\top}(\mathbf{w}_t - \mathbf{w}_{\mathcal{S}}^{*})$ to $F_{\mathcal{S}}(\mathbf{w}_t) - F_{\mathcal{S}}(\mathbf{w}_{\mathcal{S}}^{*})$

as in (5), and the value of $\mu$, i.e., the ratio of $F_{\mathcal{S}}(\mathbf{w}_t) - F_{\mathcal{S}}(\mathbf{w}_{\mathcal{S}}^*)$ to $2\|\mathbf{w}_t - \mathbf{w}_{\mathcal{S}}^*\|^2$ as in (4). For $\mathbf{w}_{\mathcal{S}}^*$, we use the solution found by stagewise SGD (V1) after a large number of iterations (200k), which gives a small objective value close to zero. We select 200 points during the process of training by stagewise SGD (V1) across all stages, and plot the curves for the values of $\theta$ and $\mu$ averaged over 5 trials in the most right panel of Figure 1. We can clearly see that our assumptions about $\mu \ll 1$ and one-point weakly quasi-convexity with $\theta > 1$ are satisfied. Hence, the provided theory for stagewise learning is applicable. We also compute the minimum eigen-value of the Hessian on several selected solutions by the Lanczos method to verify the assumption about weak convexity. The Hessian-vector product is approximated by the finite-difference using gradients. The negative value of minimum eigen-value (i.e., $\rho$) is marked as $\diamond$ in the same figure of $\theta, \mu$. We can see that the assumption about $\rho \leq O(\mu)$ seems not to hold for learning deep neural networks.

# 7 Conclusion

In this paper, we have analyzed the convergence of training error and testing error of a stagewise regularized training algorithm for solving empirical risk minimization under the Polyak-Łojasiewicz condition. We give the first theory to justify why the widely used stagewise step size scheme gives faster convergence than a polynomially decreasing step size. Our numerical experiments on deep learning verify that one class of non-convexity assumption holds and hence the provided theory of faster convergence applies. In particular, our generalization error bound analysis is based on the nice-behaved properties of non-convex functions by using uniform stability, which is designed for general non-convex problems. In the future, we consider to improve the generalization error bound for the specific conditions.

## Footnotes

[1] We can scale up $L$ such that $e^{2\hat{G}}$ is a small constant, which only scales up the bound by a constant factor.

[2]Note that here $\mu$ is not required for running the algorithm.

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
