[Supplementary Material · stagewise-learning-camera-ready-with-supplement.pdf]

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

# A. Proofs of Section 3

Theorem 2 is simply a corollary of the following theorem.

**Theorem 10.** *Suppose $\Omega = \mathbb{R}^d$, Assumption 1 holds and let $\hat{G} = G^2/L$. If $f(\cdot, \mathbf{z})$ is convex for any $\mathbf{z} \in \mathcal{Z}$, with step size $\eta_t = \frac{2t+1}{2\mu(t+1)^2}$ and $T$ iterations SGD returns a solution $\mathbf{w}_T$ satisfying*

$$\mathrm{E}_{\mathcal{A},\mathcal{S}}[F(\mathbf{w}_T)] \leq \mathrm{E}_{\mathcal{S}}[F_{\mathcal{S}}(\mathbf{w}_{\mathcal{S}}^*)] + \frac{LG^2}{2T\mu^2} + \frac{(L+2G^2)\log(T+1)}{n\mu}.$$

*If $f(\cdot, \mathbf{z})$ is non-convex for any $\mathbf{z} \in \mathcal{Z}$, with the same setting SGD returns a solution $\mathbf{w}_T$ satisfying*

$$\mathrm{E}_{\mathcal{A},\mathcal{S}}[F(\mathbf{w}_T)] \leq \mathrm{E}_{\mathcal{S}}[F_{\mathcal{S}}(\mathbf{w}_{\mathcal{S}}^*)] + \frac{LG^2}{2T\mu^2} + \frac{2T\min(2G/\sqrt{\mu}, e^{2\hat{G}})}{n-1}.$$

## A1. Proof of Theorem 10

We first present a result summarizing the uniform stability of SGD with $\Theta(1/t)$ step size.

**Theorem 11.** *Suppose Assumption 1 holds and $n > L/\mu$ is sufficiently large. If $f(\cdot, \mathbf{z})$ is convex for any $\mathbf{z} \in \mathcal{Z}$, then SGD with step size $\eta_t = \frac{2t+1}{2\mu(t+1)^2}$ satisfies uniform stability with*

$$\varepsilon_{stab} \leq \frac{L}{n\mu} + \frac{2G^2}{n\mu} \sum_{t=1}^{T} \frac{1}{t+1} \leq \frac{L + 2G^2\log(T+1)}{n\mu}.$$

*If $f(\cdot, \mathbf{z})$ is non-convex for any $\mathbf{z} \in \mathcal{Z}$, then SGD with step size $\eta_t = \frac{2t+1}{2\mu(t+1)^2}$ satisfies uniform stability with*

$$\varepsilon_{stab} \leq \frac{1+\mu/L}{n-1}(2G^2/\mu)^{1/(L/\mu+1)} T^{\frac{L/\mu}{L/\mu+1}}.$$

*Proof.* We combine the proof of Theorem 3.8 and the result of Lemma 3.11 in the long version of [14]. For applying Theorem 3.8, we need to have $\eta_t \leq 1/L$, i.e., $\frac{2t+1}{2\mu(t+1)^2} \leq 1/L$. Let us define $t_0 = \frac{L}{\mu}$. Then $\eta_t \leq 1/L, \forall t \geq t_0$. Then conditioned on $\delta_{t_0} = 0$, we apply Lemma 3.11 in [14] and have

$$\varepsilon_{stab} \leq \frac{t_0}{n} + G\mathrm{E}[\delta_T | \delta_{t_0} = 0] \leq \frac{t_0}{n} + \frac{2G^2}{n} \sum_{t=t_0}^{T} \eta_t$$

$$\leq \frac{t_0}{n} + \frac{2G^2}{n} \sum_{t=t_0}^{T} \frac{2t+1}{2\mu(t+1)^2} \leq \frac{L}{n\mu} + \frac{2G^2}{n\mu} \log(T+1). \tag{6}$$

Next we consider the case when $f(\cdot, \mathbf{z})$ is non-convex. By noting $\eta_t \leq \frac{1/\mu}{t}$, we can directly applying their Theorem 3.12 of the long version of [14] and get

$$\varepsilon_{stab} \leq \frac{1+\frac{\mu}{L}}{n-1} \left(\frac{2G^2}{\mu}\right)^{\frac{1}{1+L/\mu}} \cdot T^{\frac{L/\mu}{1+L/\mu}}.$$

$\square$

*Proof of Theorem 10.* Based on the decomposition of testing error, the result of Theorem 1 and Theorem 11, we could upper bound the testing error by combining optimization error and generalization error together. For convex problems, we have

$$\mathrm{E}_{\mathcal{A},\mathcal{S}}[F(\mathbf{w}_T)] \leq \mathrm{E}_{\mathcal{S}}[F_{\mathcal{S}}(\mathbf{w}_{\mathcal{S}}^*)] + \frac{LG^2}{2T\mu} + \frac{(L+2G^2)\log(T+1)}{n\mu}.$$

For non-convex problems, we have

$$\mathrm{E}_{\mathcal{A},\mathcal{S}}[F(\mathbf{w}_T)] \leq \mathrm{E}_{\mathcal{S}}[F_{\mathcal{S}}(\mathbf{w}_{\mathcal{S}}^*)] + \frac{LG^2}{2T\mu^2} + \frac{1+\frac{\mu}{L}}{n-1} \left(\frac{2G^2}{\mu}\right)^{\frac{1}{\frac{L}{\mu}+1}} T^{\frac{\frac{L}{\mu}}{\frac{L}{\mu}+1}}$$

$$\leq \mathrm{E}_{\mathcal{S}}[F_{\mathcal{S}}(\mathbf{w}_{\mathcal{S}}^*)] + \frac{LG^2}{2T\mu^2} + \frac{2}{n-1} \left(\frac{2G^2}{\mu}\right)^{\frac{1}{\frac{L}{\mu}+1}} T$$

Let $X = \frac{2G^2}{\mu} - 1$, which is positive when $\mu$ is very small. Given $(1+X)^{1/X} \leq e$, we have

$$\left(\frac{2G^2}{\mu}\right)^{\frac{1}{\frac{L}{\mu}+1}} = \left(\frac{2G^2}{\mu}\right)^{\frac{\mu}{2G^2-\mu} \cdot \frac{2G^2-\mu}{\mu} \cdot \frac{\mu}{L+\mu}} \leq e^{\frac{2G^2-\mu}{L+\mu}}$$

$$\leq e^{\frac{2G^2}{L}}.$$

We also have $\left(\frac{2G^2}{\mu}\right)^{\frac{1}{\frac{L}{\mu}+1}} \leq \frac{\sqrt{2}G}{\sqrt{\mu}}$ given that $\frac{2G^2}{\mu} \geq 1$ and $L/\mu \geq 1$ for small $\mu$. Thus, we complete the proof.

$\square$

# B. Proofs of Section 4

## B1. Proof of Theorem 3

We need the following lemma that states the convergence of SGD for one stage, where $E_k$ denote the conditional expectation given the randomness before $k$-th stage.

**Lemma 3.** *Suppose Assumption 1(i) and (iii) hold, and $f(\mathbf{w}, \mathbf{z})$ is a convex function of $\mathbf{w}$. By applying SGD (Algorithm 2) to $F_k = F^\gamma_{\mathbf{w}_{k-1}}$ with $\mathcal{O}(\mathbf{w}_1^k, \dots, \mathbf{w}_{T_k+1}^k) = \sum_{t=1}^{T_k} \mathbf{w}_{t+1}^k / T_k$ and $\eta \leq 1/L$, for any $\mathbf{w} \in \Omega$, we have*

$$E_k[F_k(\mathbf{w}_k) - F_k(\mathbf{w})] \leq \sigma^2 \eta_k + \frac{\|\mathbf{w}_{k-1} - \mathbf{w}\|^2}{2\eta_k T_k}.$$

*Proof.* We will prove by induction that $E[F_{\mathcal{S}}(\mathbf{w}_k) - F_{\mathcal{S}}(\mathbf{w}_{\mathcal{S}}^*)] \leq \epsilon_k$, where $\epsilon_k = \epsilon_0/2^k$, which is true for $k = 0$ by the assumption. By applying Lemma 3 to the $k$-th stage, for any $\mathbf{w}$

$$E_k[F_{\mathcal{S}}(\mathbf{w}_k) - F_{\mathcal{S}}(\mathbf{w})] \leq \frac{\|\mathbf{w}_{k-1} - \mathbf{w}\|^2}{2\gamma} + \eta_k \sigma^2 + \frac{\|\mathbf{w}_{k-1} - \mathbf{w}\|^2}{2\eta_k T_k} \qquad (7)$$

By plugging $\mathbf{w} = \mathbf{w}_{\mathcal{S}}^*$ (the closest optimal solution to $\mathbf{w}_{k-1}$) into the above inequality we have

$$E[F_{\mathcal{S}}(\mathbf{w}_k) - F(\mathbf{w}_{\mathcal{S}}^*)] \leq \frac{E[\|\mathbf{w}_{k-1} - \mathbf{w}_{\mathcal{S}}^*\|^2]}{2\gamma} + \eta_k \sigma^2 + \frac{E[\|\mathbf{w}_{k-1} - \mathbf{w}_{\mathcal{S}}^*\|^2]}{2\eta_k T_k}$$

$$\leq \frac{E[(F_{\mathcal{S}}(\mathbf{w}_{k-1}) - F_{\mathcal{S}}(\mathbf{w}_{\mathcal{S}}^*))]}{4\mu\gamma} + \eta_k \sigma^2 + \frac{E[(F_{\mathcal{S}}(\mathbf{w}_{k-1}) - F_{\mathcal{S}}(\mathbf{w}_{\mathcal{S}}^*))]}{4\mu\eta_k T_k}$$

$$\leq \frac{\epsilon_{k-1}}{4\mu\gamma} + \eta_k \sigma^2 + \frac{\epsilon_{k-1}}{4\mu\eta_k T_k}$$

where we use the result in Lemma 1. Since $\eta_k \leq \frac{\epsilon_k \alpha}{3\sigma^2}$ and $T_k \eta_k \geq 1.5/\mu$ and $\gamma_k \geq 1.5/\mu$, we have

$$E[F_{\mathcal{S}}(\mathbf{w}_k) - F_{\mathcal{S}}(\mathbf{w}_{\mathcal{S}}^*)] \leq \epsilon_k$$

By induction, after $K = \lceil \log(\epsilon_0/\epsilon) \rceil$ stages, we have

$$E[F_{\mathcal{S}}(\mathbf{w}_K) - F_{\mathcal{S}}(\mathbf{w}_{\mathcal{S}}^*)] \leq \epsilon$$

The total iteration complexity is $\sum_{k=1}^{K} T_k = O(1/(\mu\epsilon))$.

$\square$

## B2. Proof of Theorem 4

We first give the following lemma regarding the growth of stability within one stage of START. To this end, we let $f_t, f_t'$ denote the loss functions used at the $t$-th iteration of the two copies of algorithm.

**Lemma 4.** *Assume $f$ is smooth and convex. Let $\mathbf{w}_t$ denote the sequence learned on $\mathcal{S}$ and $\mathbf{w}_t'$ be the sequence learned on $\mathcal{S}'$ by START at one stage, $\delta_t = \|\mathbf{w}_t - \mathbf{w}_t'\|$. If $\eta \leq 2/L$, then*

$$\delta_{t+1} \leq \begin{cases} \frac{\eta}{\eta+\gamma}\delta_1 + \frac{\gamma}{\eta+\gamma}\delta_t & f_t = f_t' \\ \frac{\eta}{\eta+\gamma}\delta_1 + \frac{\gamma}{\eta+\gamma}\delta_t + \frac{2\eta\gamma G}{\eta+\gamma} & otherwise \end{cases}.$$

Based on the above result, we can prove the uniform stability of START.

*Proof.* By applying the result in Lemma 4 to the $k$-th stage, omitting $k$ in the notation, for $t \geq 1$ we have

$$\mathrm{E}[\delta_{t+1}] \leq (1 - 1/n)\left(\frac{\eta}{\eta + \gamma}\mathrm{E}[\delta_1] + \frac{\gamma}{\eta + \gamma}\mathrm{E}[\delta_t]\right)$$

$$+ \frac{1}{n}\left(\frac{\eta}{\eta + \gamma}\mathrm{E}[\delta_1] + \frac{\gamma}{\eta + \gamma}\mathrm{E}[\delta_t] + \frac{2\eta\gamma G}{\eta + \gamma}\right)$$

$$= \frac{\eta}{\eta + \gamma}\mathrm{E}[\delta_1] + \frac{\gamma}{\eta + \gamma}\mathrm{E}[\delta_t] + \frac{1}{n}\frac{2\eta\gamma G}{\eta + \gamma}$$

$$= \frac{\eta}{\eta + \gamma}\mathrm{E}[\delta_1]\sum_{\tau=0}^{t-1}\left(\frac{\gamma}{\eta + \gamma}\right)^{\tau} + \left(\frac{\gamma}{\eta + \gamma}\right)^{t}\mathrm{E}[\delta_1]$$

$$+ \frac{1}{n}\frac{2\eta\gamma G}{\eta + \gamma}\sum_{\tau=0}^{t-1}\left(\frac{\gamma}{\eta + \gamma}\right)^{\tau}$$

$$= \frac{\eta}{\eta + \gamma}\mathrm{E}[\delta_1]\frac{1 - (\frac{\gamma}{\eta+\gamma})^t}{1 - \frac{\gamma}{\eta+\gamma}} + \left(\frac{\gamma}{\eta + \gamma}\right)^{t}\mathrm{E}[\delta_1]$$

$$+ \frac{1}{n}\frac{2\eta\gamma G}{\eta + \gamma}\frac{1 - (\frac{\gamma}{\eta+\gamma})^t}{1 - \frac{\gamma}{\eta+\gamma}}$$

$$= \mathrm{E}[\delta_1] + \frac{2\gamma G(1 - (\frac{\gamma}{\eta+\gamma})^t)}{n}.$$

Then,

$$\mathrm{E}\left[\sum_{t=1}^{T}\delta_{t+1}/T\right] \leq \mathrm{E}[\delta_1] + \frac{2\gamma G(1 - (\frac{\gamma}{\eta+\gamma})^T)}{n}.$$

For the $k$-stage, we have $\mathbf{w}_k = \sum_{t=1}^{T}\mathbf{w}_{t+1}^k/T$ and $\mathbf{w}_{k-1} = \mathbf{w}_1$. Then

$$\mathrm{E}[\delta_k] \leq \mathrm{E}[\delta_{k-1}] + \frac{2\gamma G(1 - (\frac{\gamma}{\eta+\gamma})^{T_k})}{n},$$

where $\delta_k = \|\mathbf{w}_k - \mathbf{w}_k'\|$. By summing the above inequality for $K$ stages and noting that $\sup_{\mathbf{z}}\mathrm{E}_{\mathcal{A}}[f(\mathbf{w}_K, \mathbf{z}) - f(\mathbf{w}_K', \mathbf{z})] \leq G\|\mathbf{w}_K - \mathbf{w}_K'\|$, we prove the theorem.

□

## B3. Proof of Lemma 3

The proof of Lemma 3 follows similarly as the one of Lemma 1 in [32]. For completeness, we prove our result.

Recall that $F_k = F_{\mathcal{S}}(\mathbf{w}) + \frac{1}{2\gamma}\|\mathbf{w} - \mathbf{w}_{k-1}\|^2$. Let $r_k(\mathbf{w}) = \frac{1}{2\gamma}\|\mathbf{w} - \mathbf{w}_{k-1}\|^2 + \delta_{\Omega}(\mathbf{w})$, so $F_k(\mathbf{w}) = F_{\mathcal{S}}(\mathbf{w}) + r_k(\mathbf{w})$, where $\delta_{\Omega}(\cdot)$ is the indicator function of $\Omega$. Due to the convexity of $F_{\mathcal{S}}(\mathbf{w})$, the $\frac{1}{\gamma}$-strong convexity of $r_k(\mathbf{w})$ and the $L$-smoothness of $f(\mathbf{w}; \mathbf{z})$, we have the following three inequalities

$$F_{\mathcal{S}}(\mathbf{w}) \geq F_{\mathcal{S}}(\mathbf{w}_t) + \langle\nabla F_{\mathcal{S}}(\mathbf{w}_t), (\mathbf{w} - \mathbf{w}_t)\rangle \qquad (8)$$

$$r_k(\mathbf{w}) \geq r_k(\mathbf{w}_{t+1}) + \langle\partial r_k(\mathbf{w}_{t+1}), \mathbf{w} - \mathbf{w}_{t+1}\rangle + \frac{1}{2\gamma}\|\mathbf{w} - \mathbf{w}_{t+1}\|^2$$

$$F_{\mathcal{S}}(\mathbf{w}_t) \geq F_{\mathcal{S}}(\mathbf{w}_{t+1}) - \langle\nabla F_{\mathcal{S}}(\mathbf{w}_t), \mathbf{w}_{t+1} - \mathbf{w}_t\rangle - \frac{L}{2}\|\mathbf{w}_t - \mathbf{w}_{t+1}\|^2. \qquad (9)$$

Combining them together, we have

$$F_{\mathcal{S}}(\mathbf{w}_{t+1}) + r_k(\mathbf{w}_{t+1}) - (F_{\mathcal{S}}(\mathbf{w}) + r_k(\mathbf{w}))$$

$$\leq \langle\nabla F_{\mathcal{S}}(\mathbf{w}_t) + \partial r_k(\mathbf{w}_{t+1}), \mathbf{w}_{t+1} - \mathbf{w}\rangle + \frac{L}{2}\|\mathbf{w}_t - \mathbf{w}_{t+1}\|^2 - \frac{1}{2\gamma}\|\mathbf{w} - \mathbf{w}_{t+1}\|^2. \qquad (10)$$

Recall Line 3 of Algorithm 2, we update $\mathbf{w}_{t+1}$ as follows

$$\mathbf{w}_{t+1} = \arg\min_{\mathbf{w}\in\mathbb{R}^d}\nabla f(\mathbf{w}_t, \mathbf{z}_{i_t})^{\top}\mathbf{w} + \frac{1}{2\eta}\|\mathbf{w} - \mathbf{w}_t\|^2 + r_k(\mathbf{w}).$$

If we set the gradient of the above problem in $\mathbf{w}_{t+1}$ to 0, there exists $\partial r_k(\mathbf{w}_{t+1})$ such that

$$\partial r_k(\mathbf{w}_{t+1}) = -\nabla f(\mathbf{w}_t, \mathbf{z}_{i_t}) + \frac{1}{\eta}(\mathbf{w}_t - \mathbf{w}_{t+1}).$$

Plugging the above equation to (10), we have

$$F_{\mathcal{S}}(\mathbf{w}_{t+1}) + r_k(\mathbf{w}_{t+1}) - (F_{\mathcal{S}}(\mathbf{w}) + r_k(\mathbf{w}))$$
$$\leq \langle \nabla F_{\mathcal{S}}(\mathbf{w}_t) - \nabla f(\mathbf{w}_t, \mathbf{z}_{i_t}), \mathbf{w}_{t+1} - \mathbf{w} \rangle$$
$$+ \langle \frac{1}{\eta}(\mathbf{w}_t - \mathbf{w}_{t+1}), \mathbf{w}_{t+1} - \mathbf{w} \rangle$$
$$+ \frac{L}{2}||\mathbf{w}_t - \mathbf{w}_{t+1}||^2 - \frac{1}{2\gamma}||\mathbf{w} - \mathbf{w}_{t+1}||^2$$
$$= \langle \nabla F_{\mathcal{S}}(\mathbf{w}_t) - \nabla f(\mathbf{w}_t, \mathbf{z}_{i_t}), \mathbf{w}_{t+1} - \hat{\mathbf{w}}_{t+1} + \hat{\mathbf{w}}_{t+1} - \mathbf{w} \rangle$$
$$+ \frac{1}{2\eta}||\mathbf{w}_t - \mathbf{w}||^2 - \frac{1}{2\eta}||\mathbf{w}_t - \mathbf{w}_{t+1}||^2$$
$$- \frac{1}{2\eta}||\mathbf{w}_{t+1} - \mathbf{w}||^2 + \frac{L}{2}||\mathbf{w}_t - \mathbf{w}_{t+1}||^2 - \frac{1}{2\gamma}||\mathbf{w} - \mathbf{w}_{t+1}||^2 \quad \leq ||\nabla F_{\mathcal{S}}(\mathbf{w}_t) - \nabla f(\mathbf{w}_t, \mathbf{z}_{i_t})|| \cdot ||\mathbf{w}_{t+1} - \hat{\mathbf{w}}_{t+1}||$$
$$+ \langle \nabla F_{\mathcal{S}}(\mathbf{w}_t) - \nabla f(\mathbf{w}_t, \mathbf{z}_{i_t}), \hat{\mathbf{w}}_{t+1} - \mathbf{w} \rangle$$
$$+ \frac{1}{2\eta}||\mathbf{w}_t - \mathbf{w}||^2 - \frac{1}{2\eta}||\mathbf{w}_{t+1} - \mathbf{w}||^2 - \frac{1}{2\gamma}||\mathbf{w} - \mathbf{w}_{t+1}||^2$$
$$\leq \eta||\nabla F_{\mathcal{S}}(\mathbf{w}_t) - \nabla f(\mathbf{w}_t, \mathbf{z}_{i_t})||^2$$
$$+ \langle \nabla F_{\mathcal{S}}(\mathbf{w}_t) - \nabla f(\mathbf{w}_t, \mathbf{z}_{i_t}), \hat{\mathbf{w}}_{t+1} - \mathbf{w} \rangle$$
$$+ \frac{1}{2\eta}||\mathbf{w}_t - \mathbf{w}||^2 - \frac{1}{2\eta}||\mathbf{w}_{t+1} - \mathbf{w}||^2 - \frac{1}{2\gamma}||\mathbf{w} - \mathbf{w}_{t+1}||^2.$$

The first equality is due to

$$2\langle x - y, y - z \rangle = ||x - z||^2 - ||x - y||^2 - ||y - z||^2$$

and $\hat{\mathbf{w}}_{t+1} = \arg\min_{x \in \Omega} \mathbf{w}^\top \nabla F_{\mathcal{S}}(w) + \frac{1}{2\eta}||\mathbf{w} - \mathbf{w}_t||^2 + \frac{1}{2\gamma}||\mathbf{w} - \mathbf{w}_1||^2$. The second inequality is due to Cauchy-Schwarz inequality and setting $\eta \leq \frac{1}{L}$. The third inequality is due to Lemma 3 of [31].

Taking expectation on both sides, we have

$$\mathrm{E}[F_k(\mathbf{w}_{t+1}) - F_k(\mathbf{w})] \leq \eta\sigma^2 + \frac{1}{2\eta}||\mathbf{w}_t - \mathbf{w}||^2$$

$$- \frac{1}{2\eta}\mathrm{E}[||\mathbf{w}_{t+1} - \mathbf{w}||^2] - \frac{1}{2\gamma}\mathrm{E}[||\mathbf{w} - \mathbf{w}_{t+1}||^2],$$

where $\mathrm{E}_i[||\nabla f(\mathbf{w}, \mathbf{z}_i) - \nabla F_{\mathcal{S}}(\mathbf{w})||^2] \leq \sigma^2$ by assumption.

Taking summation of the above inequality from $t = 1$ to $T$, we have

$$\sum_{t=1}^{T} F_k(\mathbf{w}_{t+1}) - F_k(\mathbf{w}) \leq \eta\sigma^2 T + \frac{1}{2\eta}||\mathbf{w}_1 - \mathbf{w}||^2 - \frac{1}{2\eta}\mathrm{E}[||\mathbf{w}_{T+1} - \mathbf{w}||^2]$$

$$- \frac{1}{2\gamma}\sum_{t=1}^{T}\mathrm{E}[||\mathbf{w} - \mathbf{w}_{t+1}||^2].$$

By employing Jensens' inequality on LHS, denoting the output of the $k$-th stage by $\mathbf{w}_k = \hat{\mathbf{w}}_T = \frac{1}{T}\sum_{t=1}^{T}\mathbf{w}_t$ and taking expectation, we have

$$\mathrm{E}[F_k(\hat{\mathbf{w}}_T) - F_k(\mathbf{w})] \leq \sigma^2\eta + \frac{||\mathbf{w}_1 - \mathbf{w}||^2}{2\eta T}.$$

## B4. Proof of Lemma 4

*Proof.* Let us define

$$\mathcal{G}(\mathbf{u}; f, \mathbf{w}_1) = \frac{\gamma\mathbf{u} + \eta\mathbf{w}_1 - \eta\gamma\nabla f(\mathbf{u})}{\eta + \gamma}.$$

Figure 2: Illustration of $\|w - w^*\| \geq \|\nabla F(w)\|$ around a flat minimum.

It is not difficult to show that $\mathbf{w}_{t+1} = \text{Proj}_\Omega[\mathcal{G}(\mathbf{w}_t; f_t, \mathbf{w}_1)]$, where $\text{Proj}_\Omega[\cdot]$ denotes the projection operator. Due to non-expansive of the projection operator, it suffices to bound $\|G(\mathbf{w}_t; f_t, \mathbf{w}_1) - G(\mathbf{w}'_t; f'_t, \mathbf{w}'_1)\|$. Let us consider two scenarios. The first scenario is $f_t = f'_t = f$ (using the same data). Then

$$\|\mathcal{G}(\mathbf{w}_t; f, \mathbf{w}_1) - \mathcal{G}(\mathbf{w}'_t; f', \mathbf{w}'_1)\|$$
$$= \left\| \frac{\gamma \mathbf{w}_t + \eta \mathbf{w}_1 - \eta\gamma\nabla f(\mathbf{w}_t)}{\eta + \gamma} - \frac{\gamma \mathbf{w}'_t + \eta \mathbf{w}'_1 - \eta\gamma\nabla f(\mathbf{w}'_t)}{\eta + \gamma} \right\|$$
$$\leq \frac{\eta}{\eta + \gamma}\|\mathbf{w}_1 - \mathbf{w}'_1\| + \frac{\gamma}{\eta + \gamma}\|\mathbf{w}_t - \eta\nabla f(\mathbf{w}_t) - \mathbf{w}'_t + \eta\nabla f(\mathbf{w}'_t)\|$$
$$\leq \frac{\eta}{\eta + \gamma}\|\mathbf{w}_1 - \mathbf{w}'_1\| + \frac{\gamma}{\eta + \gamma}\|\mathbf{w}_t - \mathbf{w}'_t\| = \frac{\eta}{\eta + \gamma}\delta_1 + \frac{\gamma}{\eta + \gamma}\delta_t,$$

where last inequality is due to 1-expansive of GD update with $\eta \leq 2/L$ for a convex function [14]. Next, let us consider the second scenario $f_t \neq f'_t$. Then

$$\|\mathcal{G}(\mathbf{w}_t; f, \mathbf{w}_1) - \mathcal{G}(\mathbf{w}'_t; f', \mathbf{w}'_1)\|$$
$$= \left\| \frac{\gamma \mathbf{w}_t + \eta \mathbf{w}_1 - \eta\gamma\nabla f(\mathbf{w}_t)}{\eta + \gamma} - \frac{\gamma \mathbf{w}'_t + \eta \mathbf{w}'_1 - \eta\gamma\nabla f'(\mathbf{w}'_t)}{\eta + \gamma} \right\|$$
$$\leq \frac{\eta}{\eta + \gamma}\|\mathbf{w}_1 - \mathbf{w}'_1\|$$
$$\quad + \frac{\gamma}{\eta + \gamma}\|\mathbf{w}_t - \eta\nabla f(\mathbf{w}_t) - \mathbf{w}'_t + \eta\nabla f'(\mathbf{w}'_t)\|$$
$$\leq \frac{\eta}{\eta + \gamma}\|\mathbf{w}_1 - \mathbf{w}'_1\| + \frac{\gamma}{\eta + \gamma}\|\mathbf{w}_t - \mathbf{w}'_t\| + \frac{2\eta\gamma G}{\eta + \gamma}$$
$$= \frac{\eta}{\eta + \gamma}\delta_1 + \frac{\gamma}{\eta + \gamma}\delta_t + \frac{2\eta\gamma G}{\eta + \gamma}.$$

$\square$

## C. Proofs of Section 5

### C1. Proof of Lemma 2

The inequality regarding $\|\nabla F(\mathbf{w})\|^2$ can be found in [18]. The inequality regarding $\nabla F(\mathbf{w})^\top(\mathbf{w} - \mathbf{w}^*)$ can be easily seen from the definition of one-point strong convexity and the $L$-smoothness condition of $F(\mathbf{w})$ and the condition $\nabla F(\mathbf{w}^*) = 0$, i.e.,

$$F(\mathbf{w}) - F(\mathbf{w}^*) \leq \nabla F(\mathbf{w}^*)^\top(\mathbf{w} - \mathbf{w}^*) + \frac{L}{2}\|\mathbf{w} - \mathbf{w}^*\|^2$$
$$\leq \frac{L}{2\mu_1}\nabla F(\mathbf{w})^\top(\mathbf{w} - \mathbf{w}^*).$$

### C2. Proof of Theorem 6

The proof is built on the following lemma.

**Lemma 5.** *Assume $F_\mathcal{S}$ is **one-point $\theta$-weakly quasi-convex** w.r.t $\mathbf{w}_\mathcal{S}^*$. By applying SGD (Algorithm 2) to $F_k = F_{\mathbf{w}_{k-1}}^\gamma$ with $\mathbf{w}_k = \mathbf{w}_\tau$ where $\tau \in \{1, \ldots, T_k\}$ is randomly sampled, we have*

$$\mathrm{E}_k[F_\mathcal{S}(\mathbf{w}_k) - F_\mathcal{S}(\mathbf{w}_\mathcal{S}^*)] \leq \frac{\|\mathbf{w}_{k-1} - \mathbf{w}_\mathcal{S}^*\|^2}{2\theta\eta_k T_k} + \frac{\eta_k G^2}{2\theta} + \frac{1}{2\gamma\theta}\|\mathbf{w}_{k-1} - \mathbf{w}_\mathcal{S}^*\|^2$$

*Proof.* We will prove by induction that $\mathrm{E}[F(\mathbf{w}_k) - F(\mathbf{w}_*)] \leq \epsilon_k$, where $\epsilon_k = \epsilon_0/2^k$, which is true for $k = 0$ by the assumption. By applying Lemma 5 to the $k$-th stage,

$$\mathrm{E}_k[F_{\mathcal{S}}(\mathbf{w}_k) - F_{\mathcal{S}}(\mathbf{w}_{\mathcal{S}}^*)] \leq \frac{\|\mathbf{w}_{k-1} - \mathbf{w}_{\mathcal{S}}^*\|^2}{2\theta\eta_k T_k} + \frac{\eta_k G^2}{2\theta}$$

$$+ \frac{1}{2\gamma\theta}\|\mathbf{w}_{k-1} - \mathbf{w}_{\mathcal{S}}^*\|^2$$

$$\leq \frac{\mathrm{E}[(F_{\mathcal{S}}(\mathbf{w}_{k-1}) - F_{\mathcal{S}}(\mathbf{w}_{\mathcal{S}}^*))]}{4\mu\gamma\theta} + \frac{\eta_k G^2}{2\theta}$$

$$+ \frac{\mathrm{E}[(F_{\mathcal{S}}(\mathbf{w}_{k-1}) - F_{\mathcal{S}}(\mathbf{w}_{\mathcal{S}}^*))]}{4\theta\mu\eta_k T_k}$$

$$\leq \frac{\epsilon_{k-1}}{4\mu\gamma\theta} + \frac{\eta_k G^2}{2\theta} + \frac{\epsilon_{k-1}}{4\mu\theta\eta_k T_k}.$$

By the setting $\eta_k = \frac{2\epsilon_k\theta}{3G^2}$ and $T_k\eta_k = 1.5/(\theta\mu)$ and $\gamma \geq 1.5/(\theta\mu)$, we have

$$\mathrm{E}[F_{\mathcal{S}}(\mathbf{w}_k) - F_{\mathcal{S}}(\mathbf{w}_{\mathcal{S}}^*)] \leq \epsilon_k.$$

By induction, after $K = \lceil\log(\epsilon_0/\epsilon)\rceil$ stages, we have

$$\mathrm{E}[F_{\mathcal{S}}(\mathbf{w}_K) - F_{\mathcal{S}}(\mathbf{w}_{\mathcal{S}}^*)] \leq \epsilon.$$

The total iteration complexity is $\sum_{k=1}^K T_k = O(1/(\theta^2\mu\epsilon))$. $\qquad\square$

## C3. Proof of Theorem 7

The proof is built on the following lemma.

**Lemma 6.** *Assume $F_{\mathcal{S}}$ is $\rho$-weakly convex. By applying SGD (Algorithm 2) to $F_k = F_{\mathbf{w}_{k-1}}^{\gamma}$ with $\gamma \leq 1/\rho$, $\eta \leq 1/L$ and $\mathbf{w}_k = \mathcal{O}(\mathbf{w}_1^k, \dots, \mathbf{w}_{T_k+1}^k) = \sum_{t=1}^{T_k} \mathbf{w}_{t+1}^k/T_k$, for any $\mathbf{w} \in \Omega$, we have*

$$\mathrm{E}_k[F_k(\mathbf{w}_k) - F_k(\mathbf{w})] \leq \sigma^2\eta_k + \frac{\|\mathbf{w}_{k-1} - \mathbf{w}\|^2}{2T_k}\left(\frac{1}{\eta_k} + \frac{1}{\gamma}\right).$$

**Remark:** The above result indicates that $\gamma$ can not be as large as infinity. However, for a small value $\rho$, the added regularization term $1/\gamma\|\mathbf{w} - \mathbf{w}_k\|^2$ is not large.

*Proof.* We will prove by induction that $\mathrm{E}[F(\mathbf{w}_k) - F(\mathbf{w}_*)] \leq \epsilon_k$, where $\epsilon_k = \epsilon_0/2^k$, which is true for $k = 0$ by the assumption. By applying Lemma 3 to the $k$-th stage, for any $\mathbf{w} \in \Omega$

$$\mathrm{E}_k[F_{\mathcal{S}}(\mathbf{w}_k) - F_{\mathcal{S}}(\mathbf{w})] \leq \frac{\|\mathbf{w}_{k-1} - \mathbf{w}\|^2}{2\gamma} + \eta_k\sigma^2 + \frac{\|\mathbf{w}_{k-1} - \mathbf{w}\|^2}{2\eta_k T_k} + \frac{\|\mathbf{w}_{k-1} - \mathbf{w}\|^2}{2\gamma T_k}.$$

By plugging $\mathbf{w} = \mathbf{w}_{\mathcal{S}}^*$ into the above inequality we have

$$\mathrm{E}[F_{\mathcal{S}}(\mathbf{w}_k) - F_{\mathcal{S}}(\mathbf{w}_{\mathcal{S}}^*)] \leq \frac{\mathrm{E}[\|\mathbf{w}_{k-1} - \mathbf{w}_{\mathcal{S}}^*\|^2]}{2\gamma_k} + \eta_k\sigma^2 + \frac{\mathrm{E}[\|\mathbf{w}_{k-1} - \mathbf{w}_{\mathcal{S}}^*\|^2]}{2\eta_k T_k} + \frac{\mathrm{E}[\|\mathbf{w}_{k-1} - \mathbf{w}_{\mathcal{S}}^*\|^2]}{2\gamma T_k}$$

$$\leq \frac{\mathrm{E}[(F_{\mathcal{S}}(\mathbf{w}_{k-1}) - F_{\mathcal{S}}(\mathbf{w}_{\mathcal{S}}^*))]}{4\mu\gamma} + \eta_k\sigma^2 + (1/\eta_k + 1/\gamma)\frac{\mathrm{E}[(F_{\mathcal{S}}(\mathbf{w}_{k-1}) - F_{\mathcal{S}}(\mathbf{w}_{\mathcal{S}}^*))]}{4\mu T_k}$$

$$\leq \frac{\epsilon_{k-1}}{4\mu\gamma} + \eta_k\sigma^2 + (1/\eta_k + 1/\gamma)\frac{\epsilon_{k-1}}{4\mu T_k},$$

where we use Lemma 1. By the setting $\eta_k = \frac{\epsilon_k\alpha}{4\sigma^2}$ and $T_k\eta_k = 1/\mu$ and $\gamma = 4/\mu$, we have

$$\mathrm{E}[F_{\mathcal{S}}(\mathbf{w}_k) - F_{\mathcal{S}}(\mathbf{w}_{\mathcal{S}}^*)] \leq \epsilon_k.$$

By induction, after $K = \lceil\log(\epsilon_0/\epsilon)\rceil$ stages, we have

$$\mathrm{E}[F_{\mathcal{S}}(\mathbf{w}_K) - F_{\mathcal{S}}(\mathbf{w}_{\mathcal{S}}^*)] \leq \epsilon.$$

The total iteration complexity is $\sum_{k=1}^K T_k = O(1/(\mu\epsilon))$. $\qquad\square$

## C4. Proof of Lemma 5

Without loss of generality, we consider minimizing $F_1 = F_{\mathcal{S}} + \frac{1}{2\gamma}\|\mathbf{w} - \mathbf{w}_0\|^2$. Let $r(\mathbf{w}) = \frac{1}{2\gamma}\|\mathbf{w} - \mathbf{w}_0\|^2$. The initial solution of SGD $\mathbf{w}_1 = \mathbf{w}_0$. Following the standard analysis of stochastic

proximal SGD, we have

$$\nabla f(\mathbf{w}_t, \mathbf{z}_{i_t})^\top (\mathbf{w}_t - \mathbf{w}) + r(\mathbf{w}_{t+1}) - r(\mathbf{w})$$

$$\leq \frac{\|\mathbf{w} - \mathbf{w}_t\|^2}{2\eta} + \frac{\|\mathbf{w} - \mathbf{w}_{t+1}\|^2}{2\eta} + \frac{\eta}{2}\|\nabla f(\mathbf{w}_t, \mathbf{z}_{i_t})\|^2$$

Taking expectation on both sides, we have

$$\mathrm{E}[\nabla F_{\mathcal{S}}(\mathbf{w}_t)^\top (\mathbf{w}_t - \mathbf{w}) + r(\mathbf{w}_{t+1}) - r(\mathbf{w})]$$

$$\leq \mathrm{E}\left[\frac{\|\mathbf{w} - \mathbf{w}_t\|^2}{2\eta} - \frac{\|\mathbf{w} - \mathbf{w}_{t+1}\|^2}{2\eta} + \frac{\eta G^2}{2}\right]$$

Plugging $\mathbf{w} = \mathbf{w}_{\mathcal{S}}^*$, summing over $t = 1, \ldots, T$ and using the one-point weakly quasi-convexity, we have

$$\mathrm{E}\left[\sum_{t=1}^T \theta(F_{\mathcal{S}}(\mathbf{w}_t) - F_{\mathcal{S}}(\mathbf{w}_{\mathcal{S}}^*)) + r(\mathbf{w}_t) - r(\mathbf{w}_{\mathcal{S}}^*)\right] \leq \frac{\|\mathbf{w}_{\mathcal{S}}^* - \mathbf{w}_1\|^2}{2\eta} + \frac{\eta G^2 T}{2} + \mathrm{E}[r(\mathbf{w}_1) - r(\mathbf{w}_{T+1})]$$

As a result,

$$\mathrm{E}\left[F_{\mathcal{S}}(\mathbf{w}_\tau) - F_{\mathcal{S}}(\mathbf{w}_{\mathcal{S}}^*)\right] \leq \frac{\|\mathbf{w}_{\mathcal{S}}^* - \mathbf{w}_1\|^2}{2\theta\eta T} + \frac{\eta G^2}{2\theta} + \frac{1}{2\gamma\theta}\|\mathbf{w}_0 - \mathbf{w}_{\mathcal{S}}^*\|^2$$

where $\tau \in \{1, \ldots, T\}$ is randomly selected. Applying the above result to the $k$-th stage, we complete the proof.

## C5. Proof of Lemma 6

The proof of Lemma 6 follows the one of Lemma 3. The only difference lies on the weak convexity of $F_{\mathcal{S}}(\mathbf{w})$.

We could replace the first inequality in (8) by the following $\rho$-weak convexity condition of $F_{\mathcal{S}}(\cdot)$:

$$F_{\mathcal{S}}(\mathbf{w}) \geq F_{\mathcal{S}}(\mathbf{w}_t) + \langle \nabla F_{\mathcal{S}}(\mathbf{w}_t), (\mathbf{w} - \mathbf{w}_t) \rangle - \frac{\rho}{2}\|\mathbf{w}_t - \mathbf{w}\|^2.$$

Then we combine it with other two inequalities as follows

$$F_{\mathcal{S}}(\mathbf{w}_{t+1}) + r_k(\mathbf{w}_{t+1}) - (F_{\mathcal{S}}(\mathbf{w}) + r_k(\mathbf{w}))$$

$$\leq \langle \nabla F_{\mathcal{S}}(\mathbf{w}_t) + \partial r_k(\mathbf{w}_{t+1}), \mathbf{w}_{t+1} - \mathbf{w} \rangle + \frac{L}{2}\|\mathbf{w}_t - \mathbf{w}_{t+1}\|^2$$

$$- \frac{1}{2\gamma}\|\mathbf{w} - \mathbf{w}_{t+1}\|^2 + \frac{\rho}{2}\|\mathbf{w} - \mathbf{w}_t\|^2$$

Then following the proof of Lemma 3 under the condition $\eta \leq 1/L$ we have

$$F_k(\mathbf{w}_{t+1}) - F_k(\mathbf{w})$$

$$\leq \langle \nabla F_{\mathcal{S}}(\mathbf{w}_t) - \nabla f(\mathbf{w}_t, \mathbf{z}_{i_t}), \mathbf{w}_{t+1} - \hat{\mathbf{w}}_{t+1} \rangle$$

$$+ \eta\|\nabla F_{\mathcal{S}}(\mathbf{w}_t) - \nabla f(\mathbf{w}_t, \mathbf{z}_{i_t})\|^2$$

$$+ \frac{1}{2\eta}\|\mathbf{w}_t - \mathbf{w}\|^2 - \frac{1}{2\eta}\|\mathbf{w}_{t+1} - \mathbf{w}\|^2 - \frac{1}{2\gamma}\|\mathbf{w} - \mathbf{w}_{t+1}\|^2$$

$$+ \frac{\rho}{2}\|\mathbf{w} - \mathbf{w}_t\|^2$$

Taking expectation on both sides, summing from $t = 1$ to $T$ and applying Jensen's inequality, we have

$$\mathrm{E}[F_k(\hat{\mathbf{w}}_T) - F_k(\mathbf{w})] \leq \eta\sigma^2 + \frac{1}{2T\eta}\|\mathbf{w} - \mathbf{w}_1\|^2 + \frac{1}{2T\gamma}\|\mathbf{w} - \mathbf{w}_1\|^2$$

## C6. Proof of Theorem 8

We will first establish the recurrence of stability within one stage.

**Lemma 7.** *Assume $f$ is $L$-smooth. Let $\mathbf{w}_t$ denote the sequence learned on $\mathcal{S}$ and $\mathbf{w}_t'$ be the sequence learned on $\mathcal{S}'$ by* START *at one stage, $\delta_t = \|\mathbf{w}_t - \mathbf{w}_t'\|$. Then*

$$\delta_{t+1} \leq \begin{cases} \frac{\eta}{\eta+\gamma}\delta_1 + \frac{\gamma(1+\eta L)}{\eta+\gamma}\delta_t & f_t = f_t' \\ \frac{\eta}{\eta+\gamma}\delta_1 + \frac{\gamma}{\eta+\gamma}\delta_t + \frac{2\eta\gamma G}{\eta+\gamma} & otherwise \end{cases}$$

*Proof.* Let us consider two scenarios. The first scenario is $f = f'$. Then

$$\|\mathcal{G}(\mathbf{w}_t; f, \mathbf{w}_1) - \mathcal{G}(\mathbf{w}_t'; f', \mathbf{w}_1')\|$$

$$= \left\| \frac{\gamma \mathbf{w}_t + \eta \mathbf{w}_1 - \eta\gamma\nabla f(\mathbf{w}_t)}{\eta + \gamma} - \frac{\gamma \mathbf{w}_t' + \eta \mathbf{w}_1' - \eta\gamma\nabla f(\mathbf{w}_t')}{\eta + \gamma} \right\|$$

$$\leq \frac{\eta}{\eta + \gamma}\|\mathbf{w}_1 - \mathbf{w}_1'\| + \frac{\gamma(1 + \eta L)}{\eta + \gamma}\|\mathbf{w}_t - \mathbf{w}_t'\| = \frac{\eta}{\eta + \gamma}\delta_1$$

$$+ \frac{\gamma(1 + \eta L)}{\eta + \gamma}\delta_t.$$

Next, let us consider the second scenario $f \neq f'$. Then

$$\|\mathcal{G}(\mathbf{w}_t; f, \mathbf{w}_1) - \mathcal{G}(\mathbf{w}_t'; f', \mathbf{w}_1')\|$$

$$= \left\| \frac{\gamma \mathbf{w}_t + \eta \mathbf{w}_1 - \eta\gamma\nabla f(\mathbf{w}_t)}{\eta + \gamma} - \frac{\gamma \mathbf{w}_t' + \eta \mathbf{w}_1' - \eta\gamma\nabla f'(\mathbf{w}_t')}{\eta + \gamma} \right\|$$

$$\leq \frac{\eta}{\eta + \gamma}\|\mathbf{w}_1 - \mathbf{w}_1'\| + \frac{\gamma}{\eta + \gamma}\|\mathbf{w}_t - \eta\nabla f(\mathbf{w}_t) - \mathbf{w}_t' + \eta\nabla f'(\mathbf{w}_t')\|$$

$$\leq \frac{\eta}{\eta + \gamma}\|\mathbf{w}_1 - \mathbf{w}_1'\| + \frac{\gamma}{\eta + \gamma}\|\mathbf{w}_t - \mathbf{w}_t'\| + \frac{2\eta\gamma G}{\eta + \gamma}$$

$$= \frac{\eta}{\eta + \gamma}\delta_1 + \frac{\gamma}{\eta + \gamma}\delta_t + \frac{2\eta\gamma G}{\eta + \gamma}$$

$\square$

Next, we will apply the similar conditional analysis for the non-convex loss as in [14]. In particular, we will condition on $\mathbf{w}_{k-1} = \mathbf{w}_{k-1}'$, i.e., the different example will be used within the last stage, and prove the bound for $\|\mathbf{w}_K - \mathbf{w}_K'\|$. The result of Theorem 8 follows directly from Theorem 3.8 in [14] and Lemma 3.11 in the long version of [14].

### C7. Proof of Theorem 9

The proof is done simply by combining the convergence of optimization error and generalization error bound with the following simplification as done in the proof of Theorem 10:

$$\left(\frac{2G^2 c}{\mu}\right)^{\frac{1}{\frac{Lc}{\mu}+1}} = \left(\frac{2G^2 c}{\mu}\right)^{\frac{\mu}{2G^2 c - \mu} \cdot \frac{2G^2 c - \mu}{\mu} \cdot \frac{\mu}{Lc + \mu}} \leq e^{\frac{2G^2 c - \mu}{Lc + \mu}}$$

$$\leq e^{\frac{2G^2}{L}}.$$

## D. More Experimental Results

We include more experimental results in this section, including for convex loss functions and for deep learning in various settings.

### D1. Learning with Convex Loss functions

We consider minimizing an empirical loss under an $\ell_1$ norm constraint, i.e.,

$$\min_{\|\mathbf{w}\|_1 \leq B} F_{\mathcal{S}}(\mathbf{w}) = \frac{1}{n}\sum_{i=1}^{n} \ell(y_i, \mathbf{w}^\top \mathbf{x}_i),$$

where $\mathbf{x}_i \in \mathbb{R}^d$ denotes the feature vector of the $i$-th example and $y_i \in \{1, -1\}$ is the corresponding label for classification or is continuous for regression. Two loss functions are used for classification, namely squared hinge loss $\ell(y, z) = \max(0, 1 - yz)^2$ and logistic loss $\ell(y, z) = \log(1 + \exp(-yz))$. Two loss functions are also used for regression, namely square loss $\ell(y, z) = (y_i - \mathbf{w}^\top \mathbf{x}_i)^2$ and Huber loss ($\delta = 1$):

$$\ell_\delta(y, z) = \begin{cases} \frac{1}{2}(y - z)^2 & \text{if } |y - z| \leq \delta, \\ \delta(|y - z| - \frac{1}{2}\delta) & \text{otherwise,} \end{cases}$$

Figure 3: From left to right: training error, testing error, generalization error for stagewise learning with convex loss functions.

For all considered problems here, the PL condition (or the equivalent the quadratic growth condition) holds [30].

For classification, we use real-sim data from the libsvm website, which has $n = 72,309$ total examples and $d = 20,958$ features. For regression, we use the E2006-tfidf data from the libsvm website, which has $16,087$ (training examples), $3,308$ (testing examples), and $d = 150,360$ features. For real-sim data, we randomly select $10,000$ examples for testing, and for E2006, we use the provided testing set. For parameter selection, we also divide the training examples into two parts, i.e., the validation data and the training data. The size of the validation set is the same as the testing set. We run the analyzed START algorithm with $\gamma = \infty$ and the averaged solution as a returned solution at each stage. The number of iterations per-stage is determined according to the performance on the validation data, i.e., when the error on the validation data does not change significantly after 1000 iterations we terminate one stage and restart the next stage. For classification, the insignificant change means the error rate does not improve by $0.01$ and for regression it means the relative improvement on root mean square error does not change by a factor of $0.1$. The initial step sizes are tuned to get the fast training convergence and the value of $B$ are is tuned based on the performance on the validation data.

The results averaged over 5 random trails are shown in Figure 3. They clearly show the superior performance of START comparing with SGD with polynomially decaying step size.

Figure 4: From left to right: training, generalization and testing error, and verifying assumptions for stagewise learning of ResNets and ConvNet using ELU without weight decay. The results on ResNets are presented in the paper.

## D2. Non-Convex Deep Learning

We include all experimental results in this section for different settings, with differences on network structures, activation functions, with or without weight decay. The results on shown in Figure 4, 5 6, 7 with captions self-explaining the corresponding setting.

Figure 5: From left to right: training, generalization and testing error, and verifying assumptions for stagewise learning of ResNets and ConvNet using ELU with weight decay (i.e., including $5 * 10^{-4} \|\mathbf{w}\|_2^2$ regularization). The computation of $\theta$, $\mu$ and minimum eigen-value takes the regularization term into account.

Figure 6: From left to right: training, generalization and testing error, and verifying assumptions for stagewise learning of ResNets and ConvNet using RELU without weight decay. We do not compute the minimum eigen-value of the Hessian matrix because the function is non-smooth, which renders the Lanczos method based on finite-difference of gradients not converging.

Figure 7: From left to right: training, generalization and testing error, and verifying assumptions for stagewise learning of ResNets and ConvNet using RELU with weight decay (i.e., including $5 * 10^{-4}\|\mathbf{w}\|_2^2$ regularization)