[Reviews · NeurIPS 2019]

Reviewer 1



Some detailed comments are provided below: 1. It is hard to understand what Remark 6 conveys. The error bound condition (assuming that this refers to Lemma 1) is derived from the PL condition, so what it means that “this error bound condition instead of the PL condition is enough to derive the results”. 2. The analytic strategy in this paper is to first establish the convergence of SGD for each stage and then extend it to K stages for the considered START algorithm. Then, derives the test error bound of the stagewise SGD. A further step is to show how the bound can affect or guide a specific choice of K in the practical stagewise SGD. 3, It might be better to use “START” in the paper title and Figure 1, instead of “SGD”, because the regularized terms have been added to the updating formula so it is not the vanilla SGD. It would be better to discuss the link between this regularized version and vanilla SGD to further shed light on the vanilla SGD. 4, This paper mentions “Corollary 2” several times, such as in Line 183 and Line 237. However, the paper does not have Corollary 2. This paper seems to be prepared in a hurry. 5, In Line 49, definition of \mu should be given before its use. In Line 123, the assumption $| f( w, z )|\leq 1$ is strong, which only works on the function value range [-1, 1] so the authors need to give more explanation why it does not lose the generality.

Reviewer 2



This paper first performs an error decomposition of the ERM paradigm into classical statistical learning measures. These errors are bounded for the START algorithm introduced in this paper. The key concept used from statistical learning is uniform stability. While, I think this exercises is important and interesting, it was not quite clear to me which conclusion I should draw from the analysis. We know that simple algorithms, like SGD (even in non-convex cases), generalize well, so what is the main take-away message for me after reading this paper? That SGD has slow convergence is folk wisdom, so can't be the important take-away. • The terminology „testing error“ for the population risk is not very fortunate in my opinion. • Eq. 2: It should be written with respect to which variable derivatives are taken here. • If A is a randomized algorithm, what is then the meaning of f(A(S),Z)? Is A(S) a random variable, or a random probability measure (the latter point is for instance standard in game theory and also used in Shalev-Schwartz, Shamir, Srebro and Sridharan JMLR2010 „Learning, Stability and Uniform Convergence)? If you go for the second option, then there is no need to carry the heavy notation in the expectation operator. • Why is there a <= in the error decomposition after line 96? To me this looks as equality should hold. • line 119: Z_{i} can be replaced by a generic random variable, and the subscript in the expectation operator is not needed. • PL is defined here given a data sample. It is required to hold uniformly over all samples of the same length. In particular, the constant mu is deterministic. The same remark applies to the uniform bound \epsilon_{0} in line 121. • line 128: „whole“ • line 143: Expectation can be taken with respect to a generic random variable Z. There is no need for the subscript i. • I don’t understand Algorithm 2. It seems that F^{gamma}_{w} is never used in this routine? What does the function O represent at stage 5 of Algorithm 2? This becomes only clear after reading Theorem Section 4. • Line 256: “behaved”. • Line 257: I am confused by the way you want to choose theta. I assume you want theta to be very large? • Check the sentences in line 330 and 348.

Reviewer 3



The paper analyzes algorithm 1, which is very similar to the SGD with piecewise constant learning rates that people use for deep learning in practice. The theorems result from separately bounding optimization and generalization error. The optimization bound depends on one-point weak quasi-convexity (or alternatively, weak convexity), and the generalization bound is a uniform stability result. Empirical results provide evidence that these assumptions approximately hold when training ResNet models on CIFAR-10. One question I have here is why \mu (a curvature variable) is plotted as only an average \mu, unlike \theta. In cases that the model converges to approximately zero loss (most cases), the assumptions appear to be holding along the optimization trajectory. I think that at times, the paper overstates its conclusions, starting with the title. As a theory paper, it should be more careful to state the limitations in practice. Eq. 5 is an especially strong assumption for deep learning. Comparing generalization performance using uniform stability upper bounds, which are very loose, can potentially result in misleading explanations as well. Overall, I find this to be a useful step forward for theoretical understanding of deep learning optimization. Update after author response period: the authors have agreed to add more discussion of the limitations of their results. I have read through the other reviews and responses, and I will keep my review the same.

[Author Response · NeurIPS 2019]

R1: Q1: It is hard to understand what Remark 6 conveys.

A: Yes, the error bound condition refers to the inequality in Lemma 1. Lemma 1 implies that the error bound condition is weaker than the PL condition. Remark 6 means that our analysis only requires the error bound condition to hold though we assume the PL condition at the beginning since it is more widely known in the deep learning literature. The reason that we include this remark is that our experiments directly verifies the error bound condition.

R1: Q2: how the bound can affect or guide a specific choice of K in stagewise SGD.

A: The value of $K$ depends on the choice of $\epsilon$ (i.e., optimization error). Theoretically, the testing error bound (e.g., Theorem 5) allows us to find an $\epsilon$ that balances the optimization error and the generalization error. However, it only affects K in a logarithmic way. In practice, it is just a small number.

R1: Q3: It might be better to use "START" in the paper title and Figure 1, instead of "SGD".

A: Thanks for the suggestion! We will make the change. Indeed, "stagewise SGD (Vx)" are variants of START with different algorithmic choices. Their common feature is the stagewise step size scheme. stagewise SGD (V1) does not use algorithmic regularization (with $\gamma = \infty$). SGD($c/\sqrt{t}$) and SGD($c/t$) refer to the vanilla SGD (not covered by START) using two different polynomially decreasing step sizes. The comparison between stagewise SGD (Vx) and the vanilla SGD demonstrates the importance of the stagewise step size scheme, which is the key point of this paper.

R1: Q4: This paper mentions "Corollary 2" several times.

A: Sorry for the confusion. Corollary 2 should be Theorem 2.

R1: Q5: In Line 49, definition of $\mu$. In Line 123, the assumption $|f(w,z)| \leq 1$ is strong.

A: $\mu$ refers to the constant in the inequality (PL condition) just before line 49 (between line 46 and line 47). We will make it clear. The boundness assumption on $f(\mathbf{w}, \mathbf{z})$ is only used in the stability analysis for non-convex loss functions. This is following the analysis in [13]. Since a general upper bound $|f(w,z)| \leq M$ only affects the result by a constant factor, we said without loss of generality.

R2: We thank R2 for all comments.

R2: Q1: what is the main take-away message for me after reading this paper.

A: In this paper, we focus on comparing two different step size schemes instead of challenging the classical framework that either analyzes the optimization error convergence or the generalization error of SGD with a particular step size scheme. Most existing theoretical analysis of SGD uses a polynomially decreasing step size or a small step size. However, in practice people mostly use a stagewise step size for SGD, which decreases in a stagewise fashion geometrically. **The main takeaway message of this paper is that we give the first theory to justify why the widely used stagewise step size scheme gives faster convergence than a polynomially decreasing step size, i.e., the stagewise step size scheme can adapt to the nice properties of deep neural networks.** That is why we compare the results in Theorem 5 and Theorem 9 (using a stagewise step size scheme) with the result in Theorem 2 (using a polynomially decreasing step size), and in Figure 1 we compare stagewise SGD with SGD with a polynomially decreasing step size.

R2: Q2: Is $\mathcal{A}(S)$ a random variable, or a random probability measure.

A: We use $\mathcal{A}(S)$ to denote a randomized model returned by the algorithm $\mathcal{A}$ based on the dataset $\mathcal{S}$. Basically it is a random variable. Please refer to line 89. So $f(\mathcal{A}(S), Z)$ means the loss of the randomized model found by algorithm $\mathcal{A}$ on a random data $Z$.

R2: Q3: Why is there a <= in the error decomposition after line 96?

A: Yes, it is indeed an equality.

R2: Q4: It seems that $F_w^\gamma$ is never used in this routine? What does the function O represent at stage 5 of Algorithm 2?

A: We will find a better way to present it. The structure of $F_w^\gamma$ that decomposes to the original objective and a quadratic regularizer is used in Algorithm 2. The function $\mathcal{O}$ returns a solution given a sequence of intermediate solutions.

R2: Q5: Line 257: I am confused by the way you want to choose theta. I assume you want theta to be very large?

A: We expect a larger $\theta$ in order to explain the advantage of stagewise SGD compared with the vanilla SGD with a polynomially decreasing step size, since the vanilla SGD has a complexity of $O(1/(\mu^2\epsilon))$ for reaching an $\epsilon$-level of optimization error and the considered stagewise SGD has a complexity of $O(1/(\theta^2\mu\epsilon))$. Our results in Theorem 6 and Theorem 9 indicate that the larger the $\theta$, the faster the convergence of optimization error and the smaller of the testing error. Nevertheless, $\theta$ is a property of the function. A convex function has $\theta \geq 1$. Our experiments verify that for deep neural networks $\theta$ is also around 1.

Response to R3: Thanks for the comments. The green curve is actually for the $\mu$ values across all iterations (the same as $\theta$). We just add a number to mark its average value so that readers can have a sense how small is the $\mu$ as the curve is almost on zero. We will add discussion to discuss the limitation of the presented results.

[Meta-Review · NeurIPS 2019]

The paper studies an important gap in theoretical understanding of SGD vs it's practical usage, and provides interesting results. All the reviewers are positive about the paper. Congratulations on a nice result.